METHODS AND RESOURCES

# Inferring the mammal tree: Species-level sets of phylogenies for questions in ecology, evolution, and conservation

Nathan S. Upham[1,2]*, Jacob A. Esselstyn[3], Walter Jetz[1,2]*

**1** Department of Ecology & Evolutionary Biology, Yale University, New Haven, Connecticut, United States of America, **2** Center for Biodiversity & Global Change, Yale University, New Haven, Connecticut, United States of America, **3** Department of Biological Sciences and Museum of Natural Science, Louisiana State University, Baton Rouge, Louisiana, United States of America

* nathan.upham@yale.edu (NSU); walter.jetz@yale.edu (WJ)

**Data Availability Statement:** All curated data and code are available in the supplementary materials deposited in the Dryad Digital Repository: https://doi.org/10.5061/dryad.tb03d03. Code for reproducing analyses and figures is also on Github:

## Abstract

Big, time-scaled phylogenies are fundamental to connecting evolutionary processes to modern biodiversity patterns. Yet inferring reliable phylogenetic trees for thousands of species involves numerous trade-offs that have limited their utility to comparative biologists. To establish a robust evolutionary timescale for all approximately 6,000 living species of mammals, we developed credible sets of trees that capture root-to-tip uncertainty in topology and divergence times. Our "backbone-and-patch" approach to tree building applies a newly assembled 31-gene supermatrix to two levels of Bayesian inference: (1) backbone relationships and ages among major lineages, using fossil node or tip dating, and (2) species-level "patch" phylogenies with nonoverlapping in-groups that each correspond to one representative lineage in the backbone. Species unsampled for DNA are either excluded ("DNA-only" trees) or imputed within taxonomic constraints using branch lengths drawn from local birth–death models ("completed" trees). Joining time-scaled patches to backbones results in species-level trees of extant Mammalia with all branches estimated under the same modeling framework, thereby facilitating rate comparisons among lineages as disparate as marsupials and placentals. We compare our phylogenetic trees to previous estimates of mammal-wide phylogeny and divergence times, finding that (1) node ages are broadly concordant among studies, and (2) recent (tip-level) rates of speciation are estimated more accurately in our study than in previous "supertree" approaches, in which unresolved nodes led to branch-length artifacts. Credible sets of mammalian phylogenetic history are now available for download at http://vertlife.org/phylosubsets, enabling investigations of long-standing questions in comparative biology.

## Introduction

Reconstructing the timing and pattern of evolutionary relationships in the tree of life illuminates the processes of species birth (speciation), death (extinction), character evolution, and many other fundamental aspects of biodiversity generation and maintenance [1–4]. The

https://github.com/n8upham/MamPhy_v1. Credible sets of 10,000 trees are available for taxonomic subsetting using the online tool at https://vertlife.org/phylosubsets.

**Funding:** The NSF VertLife Terrestrial grant to WJ and JAE (DEB 1441737 and 1441634) and NSF grant DBI-1262600 to WJ supported this work (http://vertlife.org). The funders had no role in study design, data collection and analysis, decision to publish, or preparation of the manuscript.

**Competing interests:** The authors have declared that no competing interests exist.

**Abbreviations:** AT, adenine-thymine; bp, base pair; BLAST, Basic Local Alignment Search Tool; BAMM, Bayesian Analysis of Macroevolutionary Mixtures; BS, bootstrap support; BEAGLE, Broad-platform Evolutionary Analysis General Likelihood Evaluator; CIPRES, Cyberinfrastructure for Phylogenetic Research; ED, evolutionary distinctiveness; FBD, fossilized birth–death; GTR + G, general time-reversible plus gamma site; GTR + I + G, general time-reversible plus gamma and invariant site; Guad, Guadalupian; GC, guanine-cytosine; HPD, highest posterior density; K-Pg, Cretaceous–Paleogene; Lopi., Lopingian; Ma, million years ago; MCC, maximum clade credibility; mis-ID, misidentification; Miss., Mississippian; ML, maximum-likelihood; MRP, matrix representation parsimony; MSW2, *Mammal Species of the World, second edition*; MSW3, *Mammal Species of the World, third edition*; mtDNA, mitochondrial DNA; NCBI, National Center for Biotechnology Information; Nioge., Neogene; ND, node-dated; nDNA, nuclear DNA; Penn., Pennsylvanian; PASTIS, Phylogenetic Assembly with Soft Taxonomic Inferences; PP, posterior probability; RAxML, Randomized Axelerated Maximum Likelihood; tip DR, tip-level pure-birth diversification rate.

penchant for mammals to fossilize has made them a traditional target for studies aiming to calibrate the tempo of macroevolutionary change in global ecosystems [5–10]. Mammalian lifestyles range from subterranean burrowing to powered flight, endurance running, and even obligate marine habitation. Ecomorphological disparity accompanying fossil diversity prompted Simpson [6,11,12] to make mammals an original flagship for testing evolutionary models, including that of adaptive radiation. Of core societal relevance, mammalian phylogeny has been used to address questions of human origins [13,14], zoonotic disease outbreaks [15,16], conservation prioritization in the Anthropocene [17,18], evolutionary medicine [19,20], and the origins of ecologically important traits [21–23].

Increasingly, biodiversity questions require species-specific estimates of evolutionary processes at the tree "tips," which collectively represent the instantaneous present and probable future of biodiversity [2,24–27]. These "tip rates" [27] can either be formulated in a diversification context, reflecting the frequency of recent speciation events in a species' parent lineage (reviewed in [28]), or else in a conservation context as the extent of a species' unique evolutionary history [24,29,30]. Because the speed of recent diversification and amount of unshared evolution are roughly inverse, they offer complementary perspectives of the same information —i.e., the species-level shape of phylogenetic trees. However, for mammals and most of life, our ability to reconstruct tip rates of branching is hampered by incomplete data [31,32], as well as failures to model the error in reconstructed phylogenies with the data we do have [33,34]. Framed on a backdrop of mammalian species and population declines globally [35–37], there is clear urgency for species-level synthesis that fully accounts for estimated levels of confidence in evolutionary relationships and ages.

Therefore, in the present study, we depart from existing approaches for building consensus-based "supertrees" [38] and, instead, aim to improve the two-level approach for Bayesian estimation of "backbone-and-patch" trees that was pioneered for use in birds, squamates, and amphibians [27,39,40]. Building big phylogenies requires addressing the computational problem of how to jointly infer tree topology and branch lengths for thousands of species. Supertree approaches solve the problem by merging many small overlapping trees and, when nodes disagree, collapsing branches into polytomies (unresolved nodes) to create a "consensus" viewpoint of topology [41]. However, rate estimates derived from supertree branch lengths contain less information from the original data than rates derived from so-called supermatrix trees, in which branch lengths are inferred directly from a large matrix of characters (assuming the matrix is sufficiently complete [42] and within-matrix rate heterogeneity is modeled [43,44]).

In contrast to supertrees, the backbone-and-patch approach divides big phylogenetic problems into two nonoverlapping levels of analysis that each still computationally allow for Bayesian inference on a supermatrix of characters. These levels are (1) "backbone" divergences among major lineages (e.g., living orders and families) and (2) species-level "patch" clades with in-groups that each correspond to one representative tip on the backbone tree. Thus, the backbone and patch levels are nonoverlapping except at one shared node at the root of each patch clade (the split between in-group and out-group). To our knowledge, this two-level approach was initially proposed as a thought experiment by Mishler [45] in the context of "exemplars" and "compartments" for dividing one big computational problem into several smaller ones. It was first implemented at scale by Jetz and colleagues [27], which estimated fossil-calibrated backbone trees (two alternative topologies [46,47]) and 129 patch trees for all living birds. The approach then generates credible sets of full-sized trees (all patches plus their backbone) in a common evolutionary timescale by rescaling the relative-time patches to absolute time via the distribution of ages for the one node each patch shares on the dated backbone [27]. By comparison, the "mega-phylogeny" approach of Smith and colleagues [44] used one level of maximum-likelihood (ML) analysis to construct large consensus trees that lack a

distribution of estimated ages or relationships. Barker and colleagues [48] also used a two-level Bayesian approach to estimate an approximately 800-species phylogeny of New World nine-primaried songbirds, reinforcing the utility of dividing large computational problems into smaller nonoverlapping ones.

Herein, we describe our novel application of the backbone-and-patch approach to build a fossil-calibrated phylogeny for 5,911 living and recently extinct species of Mammalia (Fig 1). We first develop a thoroughly vetted and taxonomically reconciled DNA supermatrix for use in a global ML phylogeny, which forms a scaffold for subsetting the supermatrix into backbone- and patch-level alignments. Our goals and specific approaches are to (1) compare Bayesian node- and tip-dating strategies for fossil calibration of mammalian backbone divergences; (2) minimize the number of required monophyly assumptions when dividing the nonoverlapping levels of analysis; (3) estimate Bayesian patch clade phylogenies using a birth–death branch length prior to accommodate topological signatures of both speciation and extinction (as opposed to pure-birth models used previously [27,39,40]); and (4) thereby construct credible sets of species-level phylogenies that capture topological and branch-length uncertainty, which is then propagated to the inferred tempo of evolutionary radiation in recent Mammalia (Fig 2; see S1 Movie). These sets of phylogenetic trees are evolutionary hypotheses that provide confidence in proportion to the inferred certainty regarding mammalian divergence times and species relationships from root to tip. This is a feature designed to prevent inflated confidence in subsequent statistical tests, in which phylogenies are otherwise treated as known without error [33]. These new sets of mammal trees are available for download and subsetting, either as clades or nonmonophyletic assemblages, via an online tool: vertlife.org/phylosubsets/.

## Previous studies of Mammalia phylogeny

Most studies of mammalian evolutionary history have focused on backbone-level divergences or species-level subclade radiations, but not both. For example, Carnivora (approximately 300 living species of cats, dogs, and allies; [23,51,52]) and Cetacea (approximately 90 species of whales and dolphins; [53–56]) are particularly scrutinized because of their well-studied fossils and diverse ecological habits. At the backbone level of mammalian superordinal divergences, greater paleo- to neontological integration [57–59] has recently helped bring the "rocks and clocks" of fossil-calibrated molecular ages into greater harmony [60–65]. However, controversy persists regarding both backbone node ages (e.g., [66–72]) and topological relationships (e.g., [73–75]) despite the broad application of phylogenomic and phenotypic data. Some nodes may in fact remain obstinate (e.g., due to guanine-cytosine [GC]-biased gene conversion [76,77]). Therefore, Bayesian strategies that seek to accommodate the confidence (or lack thereof) in estimated node ages and relationships, rather than collapse it to one "best" consensus, appear most valuable for testing hypotheses related to diversification processes in mammals [33,34,59,78–80].

Only a handful of studies have ever attempted to unite species-level molecular divergences with fossil ages on a Mammalia-wide basis (Table 1). The landmark study of Bininda-Emonds and colleagues [81] used a supertree approach ("matrix representation parsimony" [MRP] [38,82]) for combining source trees estimated from either DNA or morphology into a time-scaled phylogeny of 4,510 mammal species. The MRP supertree was based on the taxonomy of *Mammal Species of the World*, *second edition* (MSW2) [83] and was updated twice: (1) Fritz and colleagues [84] linked the taxonomy to 5,020 of the 5,415 species in *Mammal Species of the World*, *third edition* (MSW3) [85] and fixed errors in the dating of bats [25,86]; and (2) Kuhn and colleagues [87] resolved the >50% of unresolved nodes (2,503 polytomies) remaining in the MRP supertree using a stochastic birth–death model, creating a set of 1,000 trees with

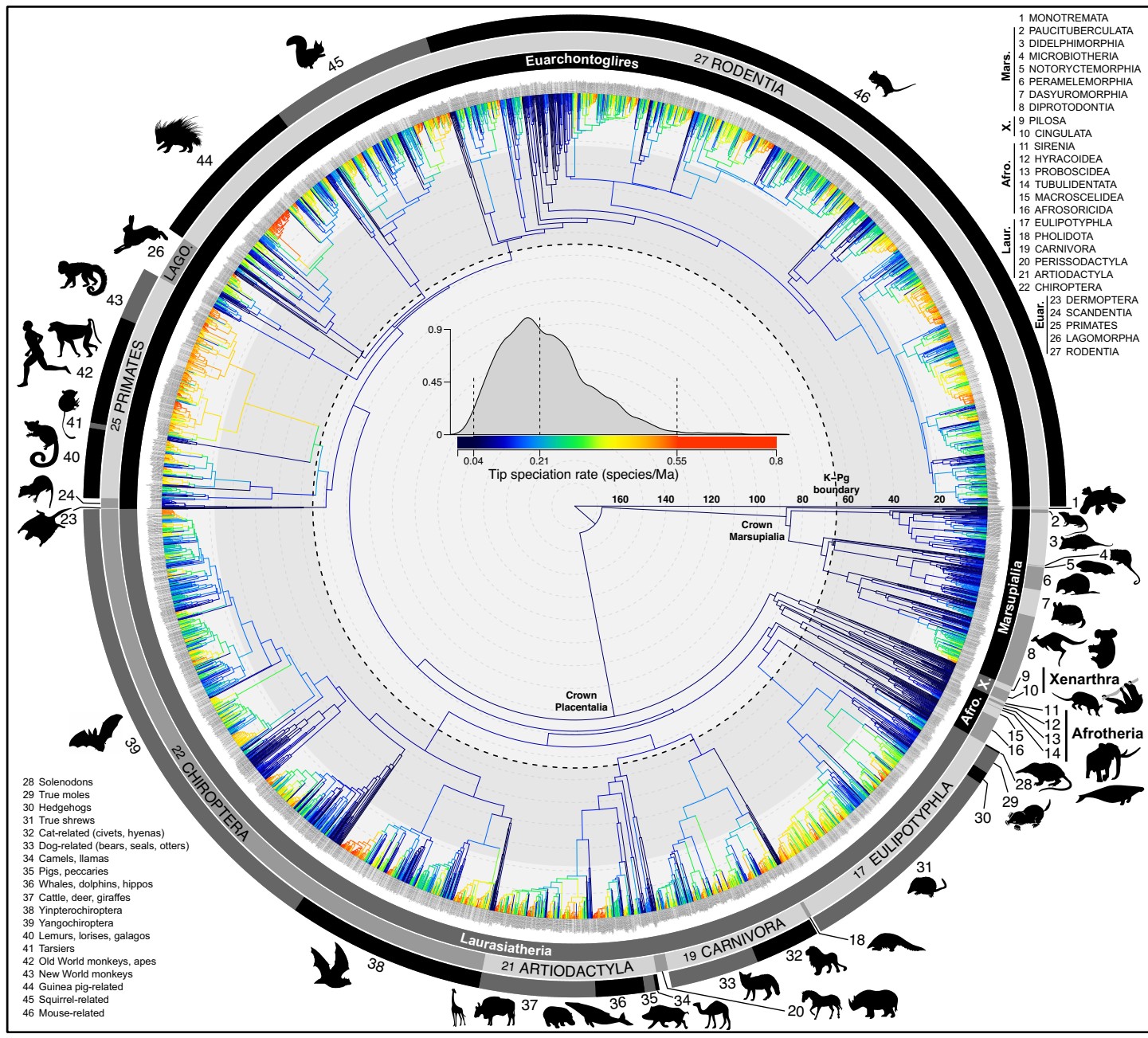

**Fig 1. Species-level relationships and tempo of diversification across mammals.** The node-dated molecular phylogeny of 5,911 extant and recently extinct species shows branches colored with tip-level speciation rates (tip DR metric; interior branches reconstructed using Brownian motion for visual purposes only). Zoom in to the branch tips to see species labels (gray branches of 1,813 species are included via taxonomic constraints rather than DNA). The maximum clade credibility topology of 10,000 trees is shown, and numbered clade labels correspond to orders and subclades listed in the plot periphery: scale in Ma. Dryad data: https://doi.org/10.5061/dryad. tb03d03; phylogeny subsets: http://vertlife.org/phylosubsets. Afro, Afrotheria; Euar, Euarchontoglires; Lago, Lagomorpha; Laur, Laurasiatheria; Ma, millions of years; Mars, Marsupialia; tip DR, tip-level pure-birth diversification rate; X, Xenarthra. *Artwork from phylopic.org and open source fonts (see S1 Text, section 9 for detailed credits).*

random variation in the placement of unresolved branches [87]. Versions of the MRP super-tree have been widely applied to questions of species diversification (e.g., [1,9,88,89]) and conservation (e.g., [25,29,90,91]) despite the initially unresolved species and consequent potential for artifacts in downstream analyses, in part because it contained the only estimates of evolutionary branch lengths across most of Mammalia.

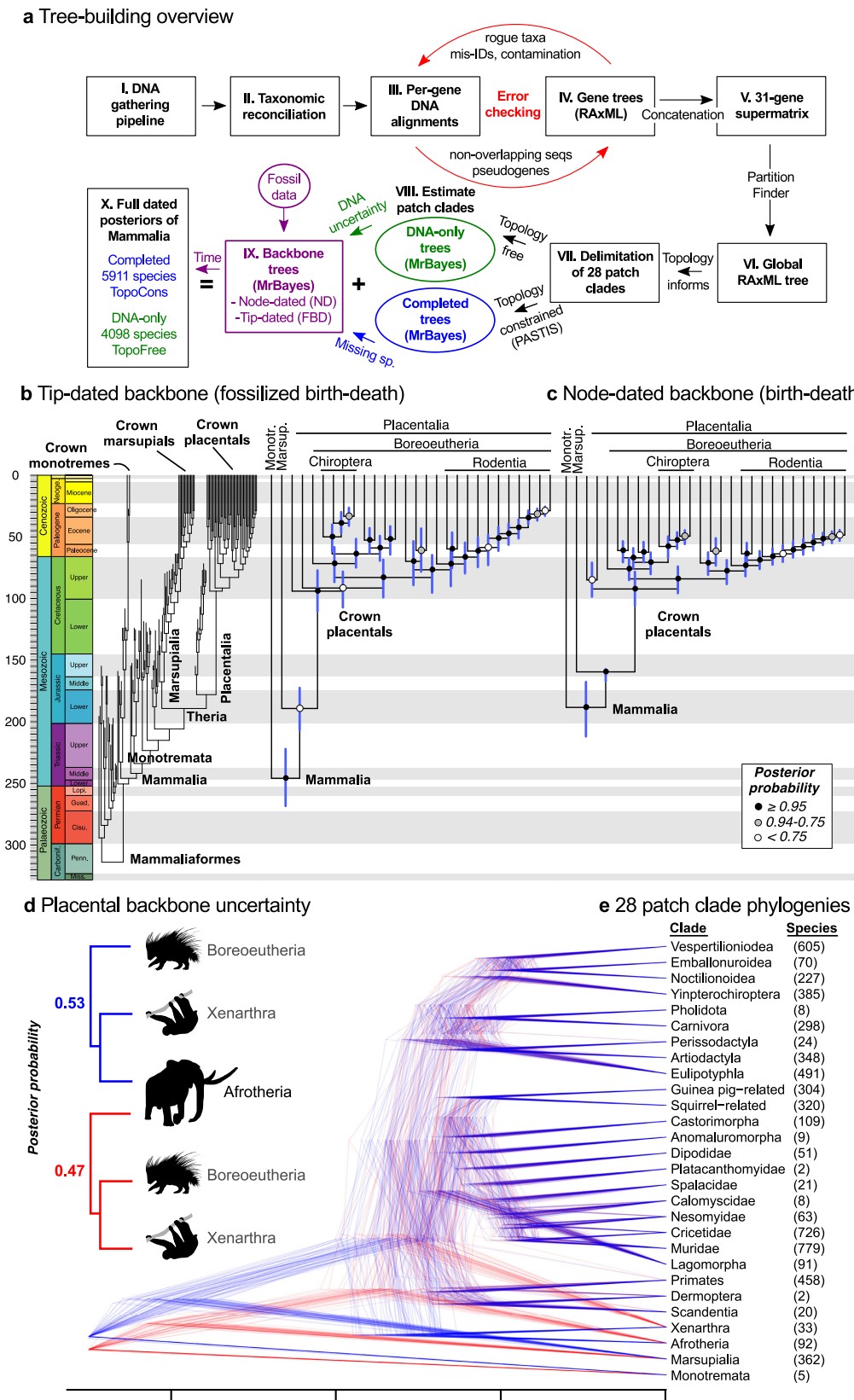

**Fig 2. Building the backbone-and-patch Mammalia phylogenies.** (a) Schematic overview of DNA sequence gathering from NCBI, taxonomic matchup, iterative error checking, and estimating a global ML tree from the resulting

supermatrix (31 genes by 4,098 species [49]). Patch phylogenies were then delimited, estimated using Bayesian inference [50], and joined to fossil-calibrated backbone trees (node- or tip-dated). The resulting posterior samples of 10,000 fully dated phylogenies either had the global ML tree topology constrained (completed trees of 5,911 species, "TopoCons") or no topology constraints (DNA-only trees, "TopoFree"). (b, c) Comparison of results from the time-calibrated backbones as pruned to the 28 patch clade representatives. The tip-dated analysis uses fossil taxa as extinct tips in the tree (left side) and then pruned (right side), whereas the node-dated approach uses exponential priors from minimum to soft-max ages. Trees are maximum clade credibility summaries of 10,000 trees. Circles at nodes indicate PP values according to the legend. (d) Topological and age uncertainty in the backbones included the unresolved base of Placentalia, which slightly favors the Atlantogenata hypothesis (blue) versus Exafroplacentalia (red; shown for the node-dated backbone). (e) Bayesian phylogenies of 28 patch clades were separately estimated in relative-time units for rescaling to representative divergence times on the backbone. Combining sets of backbones and patch clades yielded four posterior distributions for analysis. Dryad data: https://doi.org/10.5061/dryad.tb03d03; phylogeny subsets: http://vertlife.org/phylosubsets. Carbonif., Carboniferous; Cisu., Cisuralian; FBD, fossilized birth–death; Guad., Guadalupian; Lopi., Lopingian; Marsup., Marsupialia; mis-ID, misidentification; Miss., Mississippian; ML, maximum-likelihood; Monotr., Monotremata; NCBI, National Center for Biotechnology Information; Nioge., Neogene; PASTIS, Phylogenetic Assembly with Soft Taxonomic Inferences; Penn., Pennsylvanian; PP, posterior probability; RAxML, Randomized Axelerated Maximum Likelihood. *Artwork from phylopic.org and open source fonts (see S1 Text, section 9 for detailed credits).*

Only two other estimates of species-level mammalian phylogeny have been published. One was the DNA-based supertree of Faurby and Svenning [92], and the other was the consensus timetree of Hedges and colleagues [93] (Table 1). The DNA supertree was constructed from the hierarchical merging of 290 overlapping subtrees (mostly estimated at the level of genus or

**Table 1. Previous species-level phylogenies of Mammalia relative to the present study.**

| Category | Bininda-Emonds and colleagues, 2007 | Fritz and colleagues, 2009 (from Bininda-Emonds and colleagues, 2007) | Kuhn and colleagues, 2011 (from Fritz and colleagues, 2009) | Faurby and Svenning, 2015 | Hedges and colleagues, 2015 | This study |
|---|---|---|---|---|---|---|
| Tree-building method | MRP supertree | MRP supertree | MRP supertree + polytomy resolver | DNA supertree + polytomy resolver | Consensus timetree + polytomy resolver | Backbone-and-patch + PASTIS |
| Num. trees | 3* | 3* | 1,000 | 1,000 | 1$ | 10,000 (4 sets) |
| Taxonomic authority | MSW2 | MSW3 | MSW3 | IUCN + late quaternary extinct# | NCBI taxonomy | Modified IUCN + new species |
| Species | | | | | | |
| Total | 4,510 | 5,020 | 5,020 | 5,747 | 5,364 | 5,911 |
| DNA sampled | N/A | N/A | N/A | 3,383 (59%) | N/A | 4,098 (69%) |
| Extinct | 9 | 11 | 11 | 342 | 19 | 107 |
| Living | 4,501 | 5,009 | 5,009 | 5,405 | 5,343 | 5,804 |
| Genera | 1,110 | 1,188 | 1,188 | 1,371 | 1,119 | 1,283 |
| Families | 150 | 151 | 151 | 172 | 155 | 162 |
| Orders | 27 | 27 | 27 | 30 | 27 | 27 |

Variations on the supertree methodology include MRP [81,84,87], DNA-based supertrees [92], and the consensus timetree [93] using "hierarchical average linkage." MSW2 [83] and MSW3 [85] were used for the MRP supertrees, whereas IUCN [94] and NCBI [95] taxonomies were incorporated in subsequent trees. See text for discussion.

*Three trees reported with best, lower, and upper estimates for node dates.

#Extinct in the last 130,000 years, versus last approximately 500 years for the other studies.

$The "interpolated and smoothed" timetree adds missing species by genus and resolves polytomies using the Kuhn and colleagues [87] method.

Abbreviations: IUCN, International Union for the Conservation of Nature; MRP, matrix representation parsimony; MSW2, *Mammal Species of the World*, *second edition*; MSW3, *Mammal Species of the World*, *third edition*; N/A, not applicable; NCBI, National Center for Biotechnology Information; Num., number; PASTIS, Phylogenetic Assembly with Soft Taxonomic Inferences

family), followed by random resolution of polytomies, addition of DNA-lacking species
($n$ = 2,364), and subsequent rescaling to time using mean node ages from secondary (e.g., [60])
or tertiary (e.g., [81]) sources. This study was an improvement over the previous MRP super-
tree by directly estimating Bayesian subtrees using DNA sequence data (range: 1–26 markers
for most species, up to 56 for Primates), which allowed greater phylogenetic uncertainty to be
included in their final distribution of 1,000 trees. Similarly, the expansive "timetree of life" [93]
is also a supertree, albeit on the much larger scale of eukaryotes initially and then pruned to
Mammalia for application to several subsequent rate-based phylogenetic analyses (e.g., [96–
101]). Hedges and colleagues [93] merged the overlapping topologies and mean node ages
from 91 divergence-time studies of mammals published from 1991–2013 (see http://www.
timetree.org/references), using an approach they call "hierarchical average linkage" to con-
struct the supertree topology. The Hedges and colleagues [93] timetree is a single consensus
estimate (one tree). Because supertree algorithms produce polytomies when overlapping
sources disagree, the corresponding branch lengths were secondarily time scaled or simulated
in these studies as well as for the MRP supertree. Thus, evolutionary rate estimates in these
supertrees are expected to be unreliable, as the authors of the DNA supertree admit: "Our
approach places the greatest weight on the topology, which means that analyses using the result-
ing phylogeny should focus on the topology rather than on branch lengths" (see [92], page 16).

## A path forward

Reconstructing species-level mammal trees has forced researchers to depart from the standard
phylogenetic approaches for jointly inferring species relationships and node ages from primary
character data (molecular or morphological; reviewed in [102]). Steps of merging overlapping
sources, collapsing conflicting nodes, and applying point-estimate dates to scale phylogenies to
time are common to the MRP supertree, DNA supertree, and consensus timetree analyses. In
each case, branch length information is reduced from the original data as a trade-off for
enabling large-scale inference. However, with increasing computational ability and growing
public databases of DNA sequence information, supertree methods are no longer the only way
to infer big trees. We here leverage computational power from the Cyberinfrastructure for
Phylogenetic Research (CIPRES) Science Gateway [103] and extensive public data deposited
in the United States National Center for Biotechnology Information (NCBI) (Genbank [104])
to filter, clean, assemble, and then reconstruct phylogenetic history from an inclusive DNA
supermatrix of mammalian species. As we outline below, the goal of jointly inferring tree
topology and node ages is now computationally feasible for Bayesian analysis of large clades
(800–1,000 species), opening the door for greater resolution of macroevolutionary tree shape
in mammals and other taxa.

## Results

### DNA sequence alignment, gene trees, and error checking

Our BLAST-based [105] pipeline initially yielded 209,294 matching hits across all 31 genes.
We used an iterative per-gene approach to clean annotation errors in NCBI (Fig 2A), as fol-
lows: (1) sequence alignment, (2) error checking for stop codons and insufficient alignment
overlap, and (3) gene tree construction (Randomized Axelerated Maximum Likelihood
[RAxML] v.8.2.3 [49]—see S1 Text, section 3). To minimize stop codons for the 26 coding
fragments (mitochondrial DNA [mtDNA] and exons), we aligned each to the appropriate
amino acid reading frame and excluded unaligned (entirely nonoverlapping) sequences, as
well as rogue taxa (see S1 Text, section 3). In total, our error-checking steps excluded 1,618
sequences across all genes (i.e., 7.2% of the 22,504 individual DNA sequences after taxonomic

reconciliation; S2 Table). These exclusions corresponded to 119 species, yielding 4,098 species with ≥1 gene fragment validated in the final 31-gene matrix (S1 Data lists excluded sequences). Our procedure of DNA-baits searching, curation, and alignment of sequences from the NCBI database resulted in taxon sampling that ranged from 191 to 3,581 species per gene (Table 2).

### The 31-gene supermatrix

Concatenation of the per-gene alignments was performed in Geneious v.9.1 [106], resulting in a sites-by-taxon supermatrix of 39,099 base pairs (bp) and 4,098 species that was 11.9% complete in terms of ungapped sites. The final DNA supermatrix consisted of 21,021 DNA sequences from public databases (see S8 Fig for top individual contributors). We evaluated partitioning schemes for the supermatrix using PartitionFinder v.1.1.1 [107], finding that a nine-partition model was most suitable (Table 3). This model has a combined partition for APP, CREM, and FBN1 and then one partition each for BMI1; PLCB4; and first, second, and third codon partitions for nuclear DNA (nDNA) exons as well as for mtDNA fragments. For all partitions, either general time-reversible plus gamma (GTR + G) or plus gamma and invariant sites (GTR + I + G) was the best model of nucleotide evolution. We chose the simpler GTR + G model for all downstream phylogenetic analyses because including both I and G types of rate heterogeneity is known to make both model parameters difficult to estimate [49,108].

### Global RAxML tree

Phylogenetic analysis of the 4,098-species DNA matrix was first performed in RAxML with the goal to identify the single best-supported topology for mammals. For RAxML, we ran five independent analyses, each specifying 100 bootstrap replicates and using the "-f a" option and GTRCAT model to search for the best-scoring tree using ML (with this setting, the ML optimizations start from every fifth bootstrap tree [49]). Each RAxML analysis took approximately 5.7 days on 12 nodes of four threads each on the XSEDE cluster (Extreme Science and Engineering Discovery Environment; accessed via the CIPRES Science Gateway [103]). We subsequently summarized this single best ML tree (likelihood −3,383,607.6, tree length 255.3) by rooting it with *Anolis* and annotating nodes with bipartition values from 100 bootstrap replicates (S3 Data).

### Patch clade delimitation

We divided the mammalian phylogeny into 28 patch clades that were nonoverlapping in their in-group species membership (Fig 2C; Table 4). Criteria for delimitation were clade size, evidence for monophyly (in our global ML tree and previous studies), and the structure of interclade phylogenetic relationships. Nodes with >75% bootstrap support (BS) were deemed well supported. The main challenge was to balance reasonable assumptions of monophyly with maximum patch size (number of species) for which we could feasibly perform Bayesian joint estimation of topology and branch lengths in less than 1 month. If Markov chain convergence were to take longer than 1 month, the need to iteratively conduct sensitivity tests and model tuning would have been unreasonable. We used MrBayes v.3.2.6 [50] for all Bayesian inference of patch clade and backbone phylogenies because of its flexible application of topological constraints. By experimentation, we concluded that approximately 800 species was the feasibility limit for patch clade size, although matrix size and complexity also influenced run times. Near this maximum, our largest patch clade (Muridae, 778 species) took 3.7 weeks to finish 33,330,000 generations in parallel on 16 BEAGLE (Broad-platform Evolutionary Analysis General Likelihood Evaluator)-enabled compute nodes. With MrBayes run times of 1.5 to 4.5

**Table 2. Alignment details for the 31 gene fragments used in this study.**

| Gene | Region | M11 # sites | M11 # taxa | This study # sites | This study # taxa | GC content | Identical sites | Matrix comp. | Mean coverage |
|------|--------|-------------|------------|-------------------|-------------------|------------|-----------------|--------------|---------------|
| A2AB | exon | 848 | 165 (4) | 603 | 290 (A) | 61.0% | 85.2% | 91.9% | 284.1 |
| ADORA3 | exon | 332 | 163 (3) | 369 | 429 (A) | 45.8% | 83.0% | 86.6% | 427.7 |
| ADRB2 | exon | 803 | 155 (5) | 846 | 218 (A) | 53.3% | 86.1% | 94.3% | 215.1 |
| APOB | exon | 2,624 | 168 (5) | 2,523 | 587 (A) | 41.8% | 35.2% | 51.7% | 334.7 |
| APP | NC | 696 | 152 (0) | 636 | 414 (M) | 37.7% | 84.1% | 94.7% | 403.6 |
| ATP7 | exon | 686 | 163 (3) | 723 | 469 (A) | 40.6% | 83.4% | 91.5% | 459.8 |
| BCHE | exon | 995 | 149 (5) | 1,020 | 309 (A) | 40.8% | 82.0% | 94.0% | 300.3 |
| BDNF | exon | 560 | 157 (5) | 612 | 508 (A) | 55.1% | 85.3% | 87.8% | 489.4 |
| BMI1 | NC | 336 | 150 (0) | 292 | 191 (O) | 33.5% | 92.4% | 98.9% | 189.9 |
| BRCA1 | exon | 3,092 | 160 (0) | 3,264 | 969 (M) | 41.7% | 36.1% | 47.0% | 538.1 |
| BRCA2 | exon | 5,045 | 163 (0) | 4,854 | 306 (O) | 34.4% | 41.8% | 52.5% | 183.3 |
| CNR1 | exon | 1,004 | 162 (5) | 1,017 | 356 (A) | 54.0% | 83.9% | 93.8% | 342.6 |
| COI | mtDNA | -- | -- | 672 | 1,686 (A) | 43.0% | 76.6% | 96.1% | 1,657.4 |
| CREM | NC | 441 | 155 (0) | 350 | 331 (O) | 44.7% | 80.7% | 95.8% | 322.5 |
| CYTB | mtDNA | -- | -- | 1,167 | 3,581 (A) | 41.5% | 65.8% | 90.4% | 3,318 |
| DMP1 | exon | 1,361 | 161 (0) | 1,503 | 415 (M) | 51.6% | 28.3% | 41.1% | 205.3 |
| EDG1 | exon | 959 | 153 (5) | 963 | 313 (A) | 56.3% | 84.1% | 95.5% | 301.8 |
| ENAM | exon | 3,899 | 162 (0) | 3,477 | 247 (O) | 44.5% | 58.2% | 68.9% | 215.5 |
| FBN1 | NC | 745 | 150 (0) | 669 | 301 (M) | 33.6% | 81.8% | 93.6% | 289.3 |
| GHR | exon | 947 | 165 (4) | 1,044 | 978 (A) | 49.1% | 58.2% | 70.1% | 794.5 |
| IRBP | exon | 1,253 | 161 (5) | 1,146 | 1,345 (A) | 59.4% | 69.5% | 85.4% | 1,231.9 |
| ND1 | mtDNA | -- | -- | 975 | 962 (A) | 40.6% | 71.0% | 94.4% | 925.5 |
| ND2 | mtDNA | -- | -- | 1,074 | 983 (A) | 37.2% | 57.7% | 86.3% | 873.3 |
| PLCB4 | NC | 350 | 156 (0) | 288 | 478 (M) | 40.1% | 75.9% | 96.2% | 469.6 |
| PNOC | exon | 320 | 144 (0) | 339 | 410 (M) | 62.5% | 79.7% | 82.7% | 403.5 |
| RAG1a | exon | 1,799 | 168 (5) | 1,050 | 639 (A) | 51.5% | 66.2% | 84.6% | 556.8 |
| RAG1b | exon | -- | -- | 1,071 | 936 (A) | 50.6% | 61.8% | 81.1% | 763.6 |
| RAG2 | exon | 446 | 163 (5) | 450 | 889 (A) | 44.1% | 86.9% | 97.5% | 879.4 |
| TTN | exon | 4,437 | 168 (5) | 4,479 | 345 (A) | 42.0% | 41.1% | 58.9% | 205.4 |
| TYR1 | exon | 428 | 151 (5) | 429 | 336 (A) | 46.9% | 85.4% | 98.8% | 334.2 |
| VWF | exon | 1,172 | 162 (3) | 1,194 | 821 (A) | 59.0% | 64.4% | 84.0% | 707.5 |
| **Total concatenated** | | 35,603 | 164 (5) | 39,099 | 4,098 (A) | 45.3% | 13.4% | 11.9% | 1,651.2 |

Comparison is made to the family-level phylogenetic study of Meredith and colleagues [60] (M11) for the number of aligned sites (with gaps) and mammalian taxa sampled per gene. The per-gene out-group(s) are noted in parentheses, either as the number of out-group taxa used in M11 or identifying the out-group used in gene tree building for this study. Note that RAG1a and RAG1b fragments are combined in M11. Site identity is calculated pairwise, matrix comp. is the sites-by-taxa percentage of ungapped sites, and mean coverage refers to the mean ungapped sites per matrix column (i.e., mean taxon sampling per site).

Abbreviations: A, *Anolis*; comp., completeness; GC, guanine-cytosine; M, *Monodelphis*; M11, Meredith and colleagues 2011; mtDNA, mitochondrial DNA; NC, noncoding; O, *Ornithorhynchus*.

weeks for clades >200 species (Table 4), we estimate that approximately 80 weeks of run time was applied to the DNA-only and completed patch analyses for a total of 215,040 cpu hours (560 days * 24 hours * 16 nodes for final models, not counting troubleshooting).

Delimiting appropriate patch clades was especially challenging in bats and rodents. Here, species richness is highest, but so is missing genetic data and topological uncertainty. In the mouse-related clade of rodents (1,768 total species, 64% genetic sampling; [109]), we addressed

**Table 3. Model parameters optimized during the global RAxML tree search of the 4,098-species supermatrix.**

| Partition | Patterns | α | Rates | | | | | | Frequencies | | | |
|---|---|---|---|---|---|---|---|---|---|---|---|---|
| | | | A/C | A/G | A/T | C/G | C/T | G/T | A | C | G | T |
| nDNA, pos1 | 9,345 | 0.40 | 1.69 | 4.54 | 0.86 | 1.07 | 4.02 | 1.00 | 0.28 | 0.23 | 0.28 | 0.21 |
| nDNA, pos2 | 8,806 | 0.38 | 1.27 | 5.34 | 0.76 | 1.50 | 4.44 | 1.00 | 0.29 | 0.24 | 0.21 | 0.26 |
| nDNA, pos3 | 10,685 | 1.19 | 1.22 | 5.42 | 0.90 | 1.08 | 5.88 | 1.00 | 0.24 | 0.26 | 0.25 | 0.25 |
| mtDNA, pos1 | 1,253 | 0.52 | 1.08 | 6.67 | 1.70 | 0.17 | 16.01 | 1.00 | 0.30 | 0.25 | 0.22 | 0.23 |
| mtDNA, pos2 | 1,213 | 0.39 | 3.38 | 18.08 | 2.25 | 3.84 | 14.26 | 1.00 | 0.19 | 0.27 | 0.13 | 0.42 |
| mtDNA, pos3 | 1,292 | 0.05 | 0.36 | 9.53 | 0.52 | 0.46 | 6.04 | 1.00 | 0.42 | 0.32 | 0.04 | 0.22 |
| APP, CREM, FBN1 | 1,558 | 0.34 | 1.32 | 3.39 | 0.39 | 0.93 | 3.40 | 1.00 | 0.27 | 0.22 | 0.22 | 0.29 |
| BMI1 | 244 | 0.24 | 1.65 | 4.33 | 1.04 | 0.78 | 5.00 | 1.00 | 0.26 | 0.23 | 0.22 | 0.29 |
| PLCB4 | 287 | 0.49 | 1.03 | 3.67 | 0.52 | 0.78 | 3.08 | 1.00 | 0.32 | 0.24 | 0.19 | 0.25 |

Summarized per partition are the number of distinct alignment patterns, estimated alpha-shape parameter (α) of the gamma distribution, two-way rates of the GTR model of nucleotide evolution, and empirical base frequencies.

Abbreviations: GTR, general time-reversible; mtDNA, mitochondrial DNA; nDNA, nuclear DNA; pos., codon position; RAxML, Randomized Axelerated Maximum Likelihood

this issue by dividing data into two large and likely monophyletic clades (Muridae and Cricetidae) and several smaller Muroidea patch clades for which interrelationships are uncertain (Dipodidae, Spalacidae, Nesomyidae, Calomyscidae, Platacanthomyidae; [110]). We thus avoided assuming a backbone topology for mouse-related rodents; instead, uncertainty in patch interrelationships was captured on the dated Mammalia backbone (see below). Note that these smaller patches were each well supported in the global ML tree except the Nesomyidae of Madagascar (BS 72), for which monophyly is well supported in other studies [110,111].

For bats, major topological uncertainty lies within Yangochiroptera (902 species, 67% genetic sampling), especially among its most basal divergences [112–114]. However, the compute time required to run Yangochiroptera as a single patch clade was prohibitive (initial attempts suggested 6–8 weeks) and with no guarantee of convergence (matrix <10% complete). Rather, we divided this group (94 BS value, S3 Data) into three patch clades:

1. Noctilionoidea (Phyllostomidae, Mormoopidae, Noctilionidae, Thyropteridae, Furipteridae, Mystacinidae, and Myzopodidae);

2. Vespertilionoidea (Vespertilionidae, Molossidae, and Natalidae); and

3. Emballonuroidea (Emballonuridae and Nycteridae).

Most controversial of these delimitations is the placement of Myzopodidae, which we include with Noctilionoidea according to BS 76% in the global ML tree (alternatively linked to Emballonuridae [113]). Support for the Vespertilionoidea was uncertain in our global ML tree (BS 51 joining Natalidae with Molossidae + Vespertilionidae; BS 48 for Vespertilionidae with Mollosidae), and similarly, we recovered Emballonuroidea with BS 52. Nevertheless, our patch clade schema represents the best-supported hypotheses for Yangochiroptera, with diversity divided into manageable group sizes.

The remaining patch clades encompassed not only major swaths of mammalian diversity (e.g., Marsupialia, Primates) but also very small clades like Monotremata (5 species), Pholidota (8), and Dermoptera (2; Table 4). The structure of the phylogeny and backbone uncertainty necessitated small clades to minimize unsupported monophyly assumptions. Our smallest patch clades (Dermoptera and Platacanthomyidae) were needed for this reason—however, because phylogeny estimation requires at least four taxa, we added two in-group species for

**Table 4. Summary of patch clade and matrix composition.**

| # | Patch name | Total | Sample | Percent | ML tree BS | Out-group species in MrBayes | Matrix percent comp. | Patterns | Run time |
|---|---|---|---|---|---|---|---|---|---|
| 1 | Marsupialia | 362 | 279 | 77 | 97 | *Rattus norvegicus* | 18.0 | 20,617 | 4.5 |
| 2 | Afrotheria | 92 | 61 | 66 | 99 | *Dasypus novemcinctus* | 21.8 | 14,132 | <1 |
| 3 | Xenarthra | 33 | 21 | 64 | 100 | *Trichechus manatus* | 33.4 | 7,164 | <1 |
| 4 | Scandentia | 20 | 13 | 65 | 100 | *Galeopterus variegatus* | 32.5 | 3,324 | <1 |
| 5 | Primates | 458 | 333 | 73 | 95 | *Galeopterus variegatus* | 19.4 | 16,165 | 4.5 |
| 6 | Lagomorpha | 91 | 72 | 79 | 100 | *Rattus norvegicus* | 8.8 | 3,855 | <1 |
| 7 | Castorimorpha | 109 | 100 | 92 | 100 | *Rattus norvegicus* | 9.0 | 5,507 | <1 |
| 8 | Dipodidae | 51 | 33 | 16 | 100 | *Rattus norvegicus* | 15.4 | 4,166 | <1 |
| 9 | Spalacidae | 21 | 16 | 76 | 37 | *Rattus norvegicus* | 16.5 | 2,487 | <1 |
| 10 | Nesomyidae | 63 | 37 | 59 | 72 | *Rattus norvegicus* | 10.6 | 3,291 | <1 |
| 11 | Muridae | 779 | 411 | 53 | 88 | *Cricetulus barabensis* | 8.8 | 9,480 | 3.7 |
| 12 | Cricetidae | 726 | 528 | 73 | 93 | *Rattus norvegicus* | 7.3 | 9,926 | 4.5 |
| 13 | Squirrel-related | 320 | 216 | 68 | 95 | *Erethizon dorsatum* | 6.7 | 8,050 | 1.5 |
| 14 | Guinea pig–related | 304 | 204 | 67 | 100 | *Ictidomys tridecemlineatus* | 13.3 | 18,949 | 3.7 |
| 15 | Eulipotyphla | 491 | 301 | 61 | 94 | *Pteronotus parnellii* | 8.7 | 13,786 | 3.7 |
| 16 | Noctilionoidea | 227 | 190 | 84 | 76 | *Pteropus alecto* | 11.5 | 11,273 | 1.5 |
| 17 | Vespertilionoidea | 605 | 367 | 61 | 51 | *Pteropus alecto* | 7.0 | 9,694 | 3.5 |
| 18 | Emballonuroidea | 70 | 46 | 66 | 51 | *Pteropus alecto* | 9.7 | 4,013 | <1 |
| 19 | Yinpterochiroptera | 385 | 250 | 65 | 96 | *Pteronotus parnellii* | 10.6 | 11,463 | 2.5 |
| 20 | Artiodactyla | 348 | 311 | 89 | 100 | *Felis catus* | 17.3 | 14,488 | 2.0 |
| 21 | Perissodactyla | 24 | 22 | 92 | 100 | *Felis catus* | 28.1 | 3,411 | <1 |
| 22 | Carnivora | 298 | 267 | 90 | 99 | *Manis pentadactyla* | 18.0 | 12,978 | 2.2 |
| 23 | Monotremata | 5 | 3 | 60 | 100 | *Rattus norvegicus* | 47.1 | 802 | <1 |
| 24 | Pholidota | 8 | 6 | 75 | 100 | *Feliz catus* | 36.6 | 1,132 | <1 |
| 25 | Dermoptera | 2* | 2 | 100 | 100 | *Tupaia belangeri* | 79.9 | 2,094 | <1 |
| 26 | Anomaluromorpha | 9 | 4 | 44 | 100 | *Rattus norvegicus* | 46.9 | 972 | <1 |
| 27 | Calomyscidae | 8 | 3 | 38 | 100 | *Rattus norvegicus* | 23.9 | 185 | <1 |
| 28 | Platacanthomyidae | 2* | 1 | 50 | N/A | *Jaculus jaculus* | 61.5 | 1,055 | <1 |

Included are per-patch clade values for total species in our master taxonomy, number of species sampled for DNA (one or more genes), BS value for that crown group in the global RAxML tree (ML tree BS), out-group clade and designated representative in MrBayes analyses, matrix percent complete (sites-by-taxa comp. of ungapped sites, with the no-DNA species as missing data), distinct alignment patterns as determined by RAxML, and the full run time (weeks) in MrBayes if >1 week.

*Patch clades with fewer than four species were supplemented with additional related taxa for the purposes of MrBayes runs, but they were then pruned out before rescaling and pasting the patches to the backbone.

Abbreviations: BS, bootstrap support; comp., completeness; ML, maximum-likelihood; N/A, not applicable; RAxML, Randomized Axelerated Maximum Likelihood

MrBayes runs (*Callithrix jacchus* and *Gorilla gorilla*, and *Rattus norvegicus* and *Spalax ehren-bergi*, respectively). These species were pruned out before rescaling and pasting to the backbone.

We followed Jetz and colleagues [27] in classifying DNA-sampled species as type 1 (sampled for one or more genes) and DNA-missing species as type 2, 3, or 4, as follows: type 2, DNA available for at least one congeneric species (constrain to genus); type 3, no DNA in the genus, but available in the same family (constrain to family); and type 4, no DNA in the family, but available in the same order (constrain to order).

Along with 4,098 type 1 species, we had 1,649 species in the type 2 category, meaning that 91% of the 1,813 DNA-missing species could be constrained to a DNA-sampled genus. Beyond that, we had 115 genera entirely unsampled for DNA, to which 156 type 3 species belong. Most

of these missing genera are rodents (73 genera, 58 of which are muroids) or bats (22 genera). Additionally, there were three extinct families in our taxonomy to which no DNA was available at the time of download (eight species in the type 4 category): Nesophontidae, Prolagidae, and Thylacinidae (S1 Data).

## Discussion

Our mammal tree (Fig 1) traces the tempo of evolutionary history of 5,804 living and 107 recently extinct species back to the divergence of their common ancestor approximately 188 million years ago (Ma; 95% highest posterior density [HPD]: 166.7, 210.9 in the node-dated [ND] analysis). These efforts bring the evolutionary history of mammals into finer resolution and make available four credible sets of Mammalia-wide trees based on node- or tip-dated backbones and inclusion or exclusion of DNA-missing species (Fig 2). We created these phylogenetic trees as a community resource to biologists, joining an updated species-level taxonomy and a newly curated data set of 31 homologous genes for comparative analyses of molecular evolution. Critically, our synthetic effort illustrates large data gaps (e.g., approximately 30% of mammal species lack published DNA sequences). However, missing and incomplete data do not prevent the probabilistic estimation of species-level topology and branch lengths as long as phylogenetic uncertainty is treated honestly [33,115]. Philosophically, our approach aimed to minimize the false confidence associated with choosing one "best" phylogeny to represent the complex, probabilistic landscape of reconstructed macroevolutionary history (S1, S2, S3 and S4 Movies offer visual summaries of these credible sets; S9 and S10 Figs show the maximum clade credibility [MCC] consensus trees of the DNA-only data sets).

### Tip versus node dating on the mammalian backbone

Comparing our node- and tip-dating analyses, we find broadly similar backbone ages with a few exceptions (Fig 3A). Chiefly, tip dating produced an older root of Mammalia and younger divergences among some rodent and bat lineages than did the node-dating analyses (Fig 3A; S3 Table). Tip dating posits that crown mammals began radiating as early as the Permian–Triassic boundary (Fig 2B; approximately 246 Ma [222.1, 268.3]), but this is much more likely to have been an early Jurassic event [116–120]. Other recent tip-dating analyses have also recovered old ages for the Mammalia crown (e.g., approximately 204 Ma in Lee [121]), suggesting that tip dating may require a combination of root age constraints [72] and a fossilized birth–death (FBD) prior that accounts for nonrandom (diversified) taxon sampling [64] to bring estimates closer to the strict fossil age of approximately 166 Ma [66]. Here, we used the latter but not the former.

Although tip dating recovers an older root age, we find it yields younger ages than node dating for divergences between and among Muridae, Cricetidae, and Nesomyidae in the mouse-related clade and between Noctilionoidea and Emballonuroidea in Yangochiroptera bats (inset in Fig 3A; S4, S5 and S6 Figs for further details). These same areas are topologically uncertain in both backbones, indicating that the lack of monophyly constraints in the tip-dating analysis (versus 18 in the node-dating analysis) is influencing node ages. Hence, resolving the topology of difficult nodes in the rodent and bat radiations is a question deeply intertwined with resolving their divergence times. Greater applications of phylogenomic data (e.g., [122]) as well as methods that explicitly account for life-history biases among lineages (e.g., CoEvol [62]) are promising strategies toward those joint temporal and topological goals.

Overall, tip dating is laudable for its probabilistic placement of fossils using morphological synapomorphies relative to living taxa because doing so requires fewer "hard" assumptions of fossil crown-versus-stem placement [58,123]. However, for the reasons outlined above, we

**a** Backbone divergence times

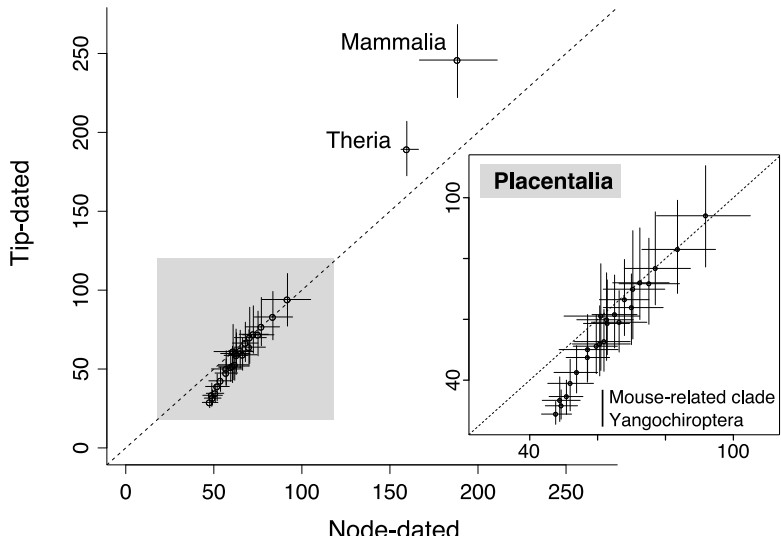

**b** Tip-level speciation rates

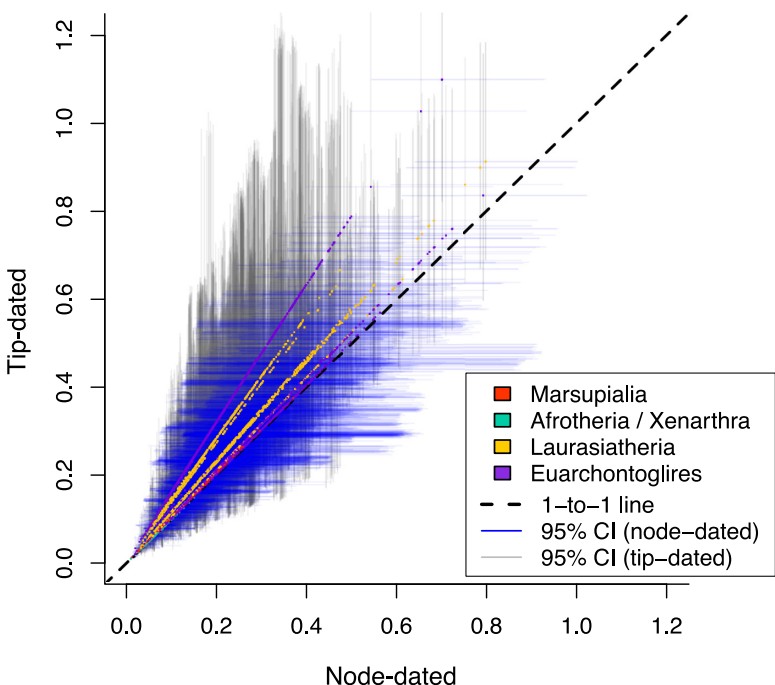

**Fig 3. Tempo of mammalian divergences recovered in the two dating analyses of the present study.** Tip-dated (fossilized birth–death) and node-dated (exponential priors) analyses yielded broadly similar results. (a) Among major clades, mean divergence times and 95% highest posterior density intervals are shown for the 28 backbone lineages present in the full trees. (b) Species-specific rates of speciation were compared using the tip DR metric, as calculated upon 10,000 trees as harmonic mean estimates (colored dots by higher taxon) and 95% CIs (Spearman's $r$ = 0.93 of tip-to node-dated harmonic means). Dryad data: https://doi.org/10.5061/dryad.tb03d03; phylogeny subsets: http://vertlife. org/phylosubsets. CI, confidence interval; tip DR, tip-level pure-birth diversification rate.

have more confidence in our node-dating analysis. The indirect use of fossil data as node priors also remains more mainstream (e.g., [69,124–128]). We thus focus discussion on how the node-dating results influence the Mammalia-wide trees relative to previous studies.

## Backbone-level divergences

Comparing our study with previous fossil-calibrated molecular trees reveals a growing consensus for the tempo of superordinal divergences in mammals (Fig 4). We find broad agreement (overlapping 95% confidence limits) for the crown age of Marsupialia in our study (approximately 79 Ma; 67.9, 92.8) relative to 68–97 Ma in previous studies (Fig 4; [60,61,81]). Similarly, for Placentalia, our crown estimate of approximately 92 Ma (77.4, 105.0) is concordant with previous studies including the tip-dating study of Ronquist and colleagues ([64]; approximately 85 Ma [76, 93]—but note an older placental age with a different tip-dating tree prior [72]; approximately 132 Ma [119, 148]). The consensus interpretation of the fossil record as given by Foley and colleagues [63] gives a wide allowance for the placental crown to be at least 65.2 Ma (*Purgatorius* stem primate [129]) and no older than 131.5 Ma (*Eomaia* stem eutherian [130]; arguably to *Juramaia* at approximately 160 Ma [117]). Nevertheless, the strict fossil-based perspective for marsupial and placental crown ages fixed at 64.85 Ma [66] appears untenable given joint consideration of the molecular and fossil evidence.

Particular controversies exist regarding whether early divergences in crown placentals occurred before, after, or during the Cretaceous–Paleogene (K-Pg) mass extinction event, 66 Ma [131]. Here, we recover the first four placental divergences unambiguously preceding the K-Pg: (1) Atlantogenata (when present; see below), (2) Boreoeutheria, (3) Laurasiatheria, and (4) Euarchontoglires (Fig 4). The next 21 divergence events subsequently have confidence limits that overlap the K-Pg, including 12 superordinal divergences and nine of 18 crown orders (Fig 4; S3 Table). The K-Pg event being possibly concurrent with nine of the 18 placental orders compares with previous studies finding three [61], five [81], or six [60] orders with K-Pg-overlapping divergence times (S3 Table). Our finding that no placental crown ordinal radiation definitively preceded approximately 66 Ma counters previous evidence that Eulipotyphla [60,81] and possibly Rodentia and Primates [64,81] began radiating before this event (Fig 4).

Our node-dating results are conservative with respect to the fossil record. Considering the oldest fossil genus per extant mammalian order (yellow squares in Fig 4; data from the Paleobiology Database [132]), we found consistent agreement with the expectation for these fossils to be members of the stem lineage and thus older than the crown ages from the phylogeny. These maximum fossil ages, when available, are found to be either older or overlapping our divergence-time intervals in all but two cases (Fig 4). These exceptions are (1) Diprotodontia, in which the fossil genus *Paljara* (Pseudocheiridae) may be as old as approximately 34 Ma [133] versus approximately 49 Ma for the crown order in our tree; and (2) Eulipotyphla, in which *Litolestes* and *Oncocherus* are Erinaceidae from as old as approximately 62 Ma [134] versus approximately 75 Ma in our tree. In both cases, molecular ages extend back further, suggesting that those fossils are either legitimate crown rather than stem members of those orders or are later-surviving stem representatives. The former case is supported for *Litolestes* ([66]; S4 Table compares these fossils to our stem ages).

An additional check relative to the fossil record was to search for "zombie lineages" [125], in which molecular divergence dates are younger (more recent) than the minimum ages implied by well-supported crown fossils. Comparing our dates with the consensus node calibrations of Foley and colleagues ([63] updated from those of [60]), we find broad agreement but three notable exceptions (asterisks on taxon names in Fig 4). First, and most substantially,

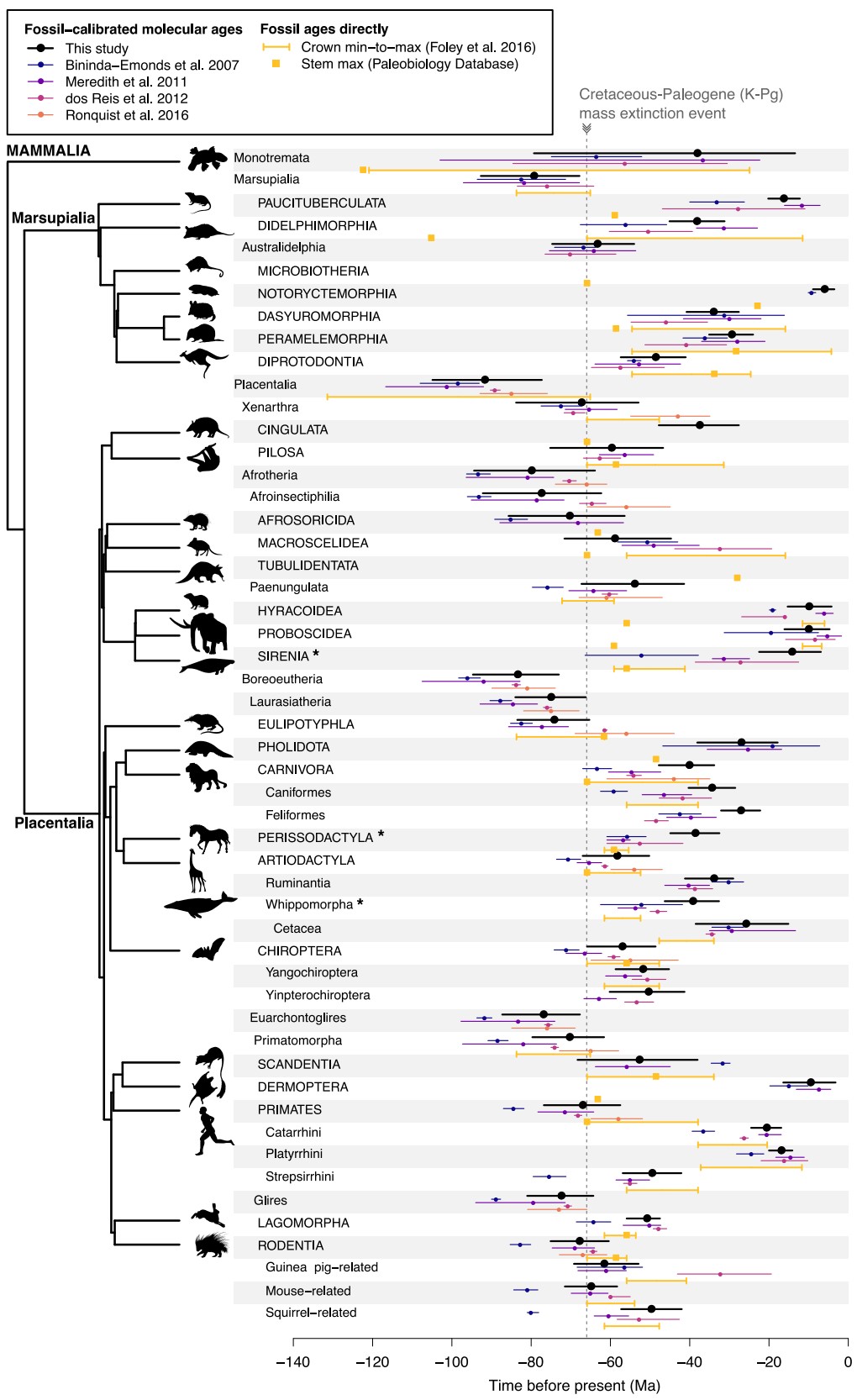

**Fig 4. Mammalian backbone-level divergence times in our study (node-dated analysis) versus previous studies and the fossil record.** The right-side phylogeny depicts relationships among the 27 extant orders (labeled in capital letters and nested in a hierarchical list), and the dotted line represents the K-Pg extinction event, 66 Ma. Divergence times are colored per study as mean ages and 95% confidence intervals. Fossil-calibrated molecular ages are compared with min and max ages for the oldest crown fossil according to Foley and colleagues [63] and oldest stem fossil according to the Paleobiology Database. Asterisks (*) on taxon names denote three instances of "zombie lineage" disagreement of our study with previous interpretations of the fossil record (see Discussion). Note that extant Microbiotheria and Tubulidentata are monotypic, and so they lack crown ages. Dryad data: https://doi.org/10.5061/dryad.tb03d03; phylogeny subsets: http://vertlife.org/phylosubsets. K-Pg, Cretaceous–Paleogene; Ma, million years ago; max, maximum; min, minimum. *Artwork from phylopic.org and open source fonts (see S1 Text, section 9 for detailed credits).*

our crown age for Sirenia (manatees, dugongs, and sea cows; approximately 13 Ma: 7.0, 22.6) is reconstructed as younger than the minimum age constraint of 41.3 Ma given in Foley and colleagues [63] and, thus, "undead" for at least 20 Ma. We reconstruct the sirenian stem divergence at approximately 54 Ma (41.5, 67.3), which implies a long stem to the crown divergence of *Dugong*, *Trichechus*, and *Hydrodamalis*—rather than the perspective in which those three modern genera are deeply divergent from each other [60,63,135]. This issue hinges on the acceptance of the fossils *Halitherium* and *Eotheroides* as crown sirenians because they form the minimum age constraint in previous studies. The most recent cladistic analysis of Springer and colleagues [135] found 40% BS for the placement of *Halitherium* and *Eotheroides* inside of crown Sirenia (stem taxa of the *Dugong*–*Hydrodamalis* clade, to the exclusion of *Trichechus* [135]). Based on our criteria for fossil inclusion [57], these fossils were placed too tenuously for use as Sirenia crown constraints. Instead, we relied on a single node prior for Afrotheria (calibration 7 in S1 Text) and, thereby, placed greater weight on molecular evidence for this node. Philosophically, we recognize that we set a high bar for placing fossils using cladistic analyses, but we contend this approach is necessary to avoid false confidence regarding the timescale of mammalian evolution.

Second, molecular divergences within the sister clades Perissodactyla and Artiodactyla in our analyses also display apparent zombie tendencies relative to some interpretations of the fossil record (Fig 4). The crown or stem placement of constraint fossils is again in question. We recover the crown divergence of Perissodactyla at approximately 39 Ma (32.6, 45.0), which overlaps the age estimate obtained in dos Reis and colleagues [61] of 52.6 Ma (41.8, 61.0) and is similar to the mean age estimates of Phillips [126] of 41.4 Ma and 36.1 Ma based on strict and relaxed molecular clocks. In contrast, calibrating Perissodactyla with the fossil genus *Hyracotherium* sets an age range of 55.5–61.6 Ma [60,63], which was closely mirrored in the Perissodactyla age of 56.8 Ma (55.1, 61) recovered in Meredith and colleagues ([60]; Fig 4). However, the only two cladistic studies of *Hyracotherium* show it falling outside of crown Perissodactlya: O'Leary and Gatesy [136] and Spaulding and colleagues [137]. Both studies recovered *Hyracotherium* as stemward to the clade that includes crown Perissodactlya + Artiodactyla; therefore, this fossil is actually two nodes back from being able to serve as a crown Perissodactlya constraint. The next candidate fossil for the oldest crown Perissodactlya is younger than the fossil we used to calibrate Artiodactyla: *Himalayacetus subathuensis* from the early Eocene approximately 52.4 Ma of India, which is the oldest stem whale according to the cladistic analysis of O'Leary and Uhen [138] (calibration 16 in S1 Text, following the compendium of Benton and colleagues [139]).

Our use of *Himalayacetus* to calibrate Artiodactlya, in turn, informs our recovered age of approximately 39 Ma (32.7, 46.4) for crown Whippomorpha (whales + hippos). Foley and colleagues [63] use that same fossil as a crown constraint for the Whippomorpha node (52.5–61.6 Ma), which is three full nodes tipward from the Artiodactyla crown, where we used it. Clearly, this is another case of differently interpreting the fossil record. *Himalayacetus* is known only

from a partial dentary and two molars [140] and is tentatively allied as a stem whale [66] but is more conservatively a stem whippomorphan for use in calibrating the Ruminantia–Whippomorpha node as Benton and colleagues [139] recommend (we did not do this to avoid using the same fossil twice). Although our crown ages for Whippomorpha and Cetacea are somewhat young, the former overlaps previous age estimates of 52.2 Ma (41.9, 62.6; [81]) and 48.1 Ma (45.9, 50.1; [61]), and the latter is congruent with three previous studies (see Fig 4).

In summary, interpretations of the fossil record that lose sight of the need to cladistically confirm the placement of calibration fossils inside the crown node they are constraining appear to cause the three putative zombie disagreements between our study and Foley and colleagues [63]. Our more conservative application of the fossil record aims to exclude opinion-based assignments of fossils in crown clades [57]. The fossil record provides critical data about past mammalian diversity [11,141,142], but because the preservation of mammal fossils is spatially, temporally, and taxonomically biased (e.g., Cenozoic of North America [143]; bats [144]), we contend that providing greater weight to the molecular data is warranted. It is possible to place a priori constraints on the ages of nearly all nodes in the mammalian backbone (e.g., 84 of the 163 nodes in Meredith and colleagues [60]). However, doing so requires the use of taxonomic opinions to place fossils relative to given crown groups. As we detailed above, some of these opinion-placed fossils are subsequently found to be stemward of the calibrated node in cladistic analyses. Because both "clocks" and "rocks" have shortfalls [59,145], the implementation of fossil calibration approaches to molecular data should aim to propagate age uncertainty rather than overly restrict it, thereby enabling conservative tests of evolutionary history and its causal underpinnings.

## Backbone-level topology

The objective of our study was to provide novel resolution on the rates and timing of mammalian divergence events, but these results are nevertheless relevant to a few long-standing issues of topological relationships among major clades (see S1 Text and S4, S5 and S6 Figs for detailed comparisons of the backbone consensus trees). We highlight four regions of the placental backbone that are especially controversial:

1. The rooting of Placentalia. We recover support of 0.53 posterior probability (PP) in favor of the Atlantogenata rooting (Xenarthra + Afrotheria) compared with 0.47 PP for the Afrotheria rooting (Exafroplacentalia) in the ND analyses (Fig 2B and 2C), whereas the tip-dated backbone recovered the Afrotheria rooting most commonly (0.44 PP; rooting of Atlantogenata was also recovered). The high uncertainty we recover for this basal divergence is typical of other molecular studies [60,61,63,73,75,128], although the Atlantogenata rooting has received more support in phylogenomic data sets (e.g., [75]). In contrast, studies that filter genes based on their likelihood of incomplete lineage sorting (proxied by adenine-thymine [AT] content; [73,77]) generally favor the Afrotheria rooting.

2. The position of treeshrews (Scandentia) relative to colugos (Dermoptera) and Primates. We find treeshrews allied with colugos (0.78 and 0.84 PP in node- and tip-dated analyses), and that clade is always adjacent to Primates. By comparison, Scandentia has varied in position considerably depending on analysis methodology in other studies, mostly between the result we recovered and rooting outside all other Euarchontoglires (including rodents and lagomorphs; e.g., [73,75,128]).

3. The position of guinea pig–related rodents (Hystricomorpha, also called Ctenohystrica; see [146]) relative to mouse- and squirrel-related clades. We find this controversial node, which was formerly questioned to even be inside Rodentia [147], to be unequivocally

recovered as ([guinea pig, squirrel] mouse) in all backbone analyses. Strong support for this relationship was recovered in some studies [60,75,122], but others have supported squirrels outside other rodents, either with a total-evidence approach [148] or when taxon sampling is smaller [61,73,128]. Transposon evidence suggests that ancient hybridization may be complicating the early history of rodents [109], as might the disparate rates of molecular evolution in these three clades [122]. Regardless of the order of branching, these basal rodent divergences were very rapid and possibly even simultaneous (i.e., overlapping error bars for nodes 45–47 in S6 Fig).

4. The branching order of mouse-related rodent families (backbone of Supramyomorpha [146]). We recover the infraorder Myomorphi as marginally sister to Anomaluromorphi to the exclusion of Castorimorphi (0.52 PP in the node-dating backbone), as well as Muridae–Cricetidae to the exclusion of Nesomyidae somewhat favored (0.57 PP; Fig 2B). These nodes were also equivocal in the largest phylogenomic data set yet leveled at this question [122], and the study of Steppan and Schenk ([110]; six loci, 904 muroid taxa) found 93% ML BS for the Muridae–Cricetidae relationship. Again, these interfamilial mouse-related divergences appear to have been extremely rapid.

We emphasize the retained uncertainty in the placental backbone divergence (Fig 2C) as a strength of the backbone-and-patch approach because having two levels of nonoverlapping Bayesian analysis enables temporal information to be passed forward to the species tips. Rather than selecting one "best" topology for rooting the placental radiation, our trees propagate the implications of both Atlantogenata and Afrotheria rootings (and other uncertainties) to the final species-level sets of 10,000 trees, providing investigators with a more realistic resource for hypothesis testing than any single tree alone.

## Species-level tree shape

Comparing the accumulation of lineages through time demonstrates how the temporal and phylogenetic uncertainty propagated in our mammal trees surpasses that of previous species-level studies (Fig 5). Viewing these phylogenies on a Mammalia-wide basis (Fig 5A), we see that the range of ages incorporated in the original MRP supertree [81,84], and subsequent analyses that resolved polytomies [87], is considerably narrower than the ages encompassed in our credible tree sets. The large number of polytomies originally present in the MRP supertree is shown concentrated at approximately 50 Ma and approximately 30 Ma (Fig 5A), particularly within the rodents (Fig 5B, lower right, top line). Resolving those polytomies changes the tree shape but does not reflect the considerable uncertainty in node ages and relationships. That is, the unresolved nodes produced in supertree studies when nodes conflict are "soft" polytomies, in which the data needed to resolve a given node is lacking [41], as opposed to "hard" poly-tomies, in which historically rapid divergence has led to a star phylogeny [149]. Collapsing uncertainty into soft polytomies was a purposeful tool for supertree methods to yield a single consensus picture of evolutionary topology for more species than possible under joint inference [38,41,150].

The danger, of course, has been when soft polytomies are misinterpreted by subsequent investigators who assume that all temporal and phylogenetic signatures in supertrees are driven by biological processes. For example, the study of Stadler and colleagues [88] made an important modeling advance for detecting tree-wide shifts in diversification rates, but the bio-logical conclusion of a major rate shift approximately 30 Ma in rodents was apparently driven by soft polytomies in the MRP supertree (see Fig 5B). Miscommunication between the stated purpose of supertrees—"to produce phylogenies based on all data sources" [150: 266]—and

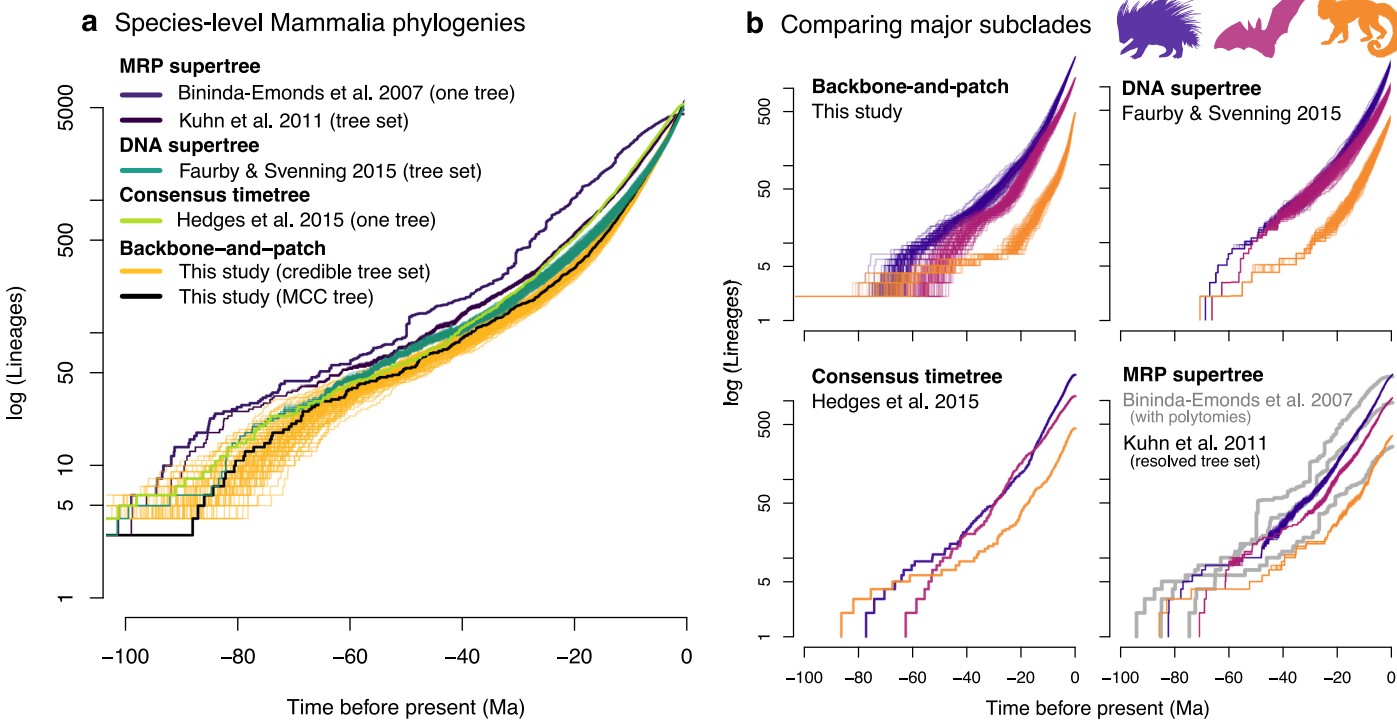

**Fig 5. Accumulation of lineages through time for living mammal species as reconstructed in our study versus previous studies.** (a) The shape of Mammalia-wide phylogenies is compared among studies using the natural log of lineage accumulation (see legend colors). Some studies produced one consensus tree (single line), whereas other studies produced sets of 1,000 or 10,000 trees (many lines), in which case 100 trees were randomly sampled. (b) Each of the main species-level Mammalia studies is compared for three major placental orders: Rodentia (purple), Chiroptera (red), and Primates (orange). The degrees of phylogenetic uncertainty present in the tree sets is represented by the width of the lineage accumulation curves. The gray lines in the lower-right-side plot pertain to the MRP supertree with polytomies, whereas the colored lines result from randomly resolving those polytomies into a set of 1,000 trees, of which 100 trees are plotted here. Dryad data: https://doi.org/10.5061/dryad.tb03d03; phylogeny subsets: http://vertlife.org/phylosubsets. Ma, million years ago; MCC, maximum clade credibility; MRP, matrix representation parsimony. *Artwork from phylopic.org and open source fonts (see S1 Text, section 9 for detailed credits).*

the need for big trees to additionally model all uncertainty in those data sources appears to have limited the durability of supertree-based inferences and, perhaps, non-Bayesian methods generally [33].

The other mammal supertrees similarly contain less temporal uncertainty in their lineage accumulation curves than the backbone-and-patch trees of this study (Fig 5). Constructed directly from genus- and family-level DNA trees, the DNA supertree study of Faurby and Svenning [92] represents an advance over the MRP supertree. However, as shown in the lineage accumulation curves of rodents, bats, and primates (Fig 5B, top right), there are unusual artifacts of limited temporal uncertainty between those crown ordinal divergences until approximately 55 Ma, when the curves broaden to represent greater rate uncertainty. Are paleomammalogists actually more certain about the timing of events near the K-Pg extinction event 66 Ma than they are about modern divergences? Although this seems unlikely given preservation biases in the fossil record (e.g., [151,152]), that is the information conveyed by the DNA supertree. However, rather than an intended statement of confidence by the study's authors, this is an artifact of the hierarchical merging and rescaling of overlapping subtrees onto a time-scaled backbone that lacks age uncertainty. Although the authors of the DNA supertree contend that applications "should focus on the topology rather than on branch lengths" (see [92], page 16), that advice may be subsequently ignored by researchers because of

the allure of addressing important questions in comparative biology. Parsimony-based ancestral state reconstruction is perhaps the only methodological approach that entirely neglects branch lengths [102,153]. In this context, nearly all phylogenetic questions are to some extent "rate based," although the relative importance of tree- and rate-based information to different questions is subject to debate. The key point is that researchers seeking to perform all but the most basic parsimony analyses are aided by phylogenies that propagate uncertainty in both rates and topology.

### Tip-level speciation rates of mammals

We calculated tip-level speciation rates across all living mammal species (Fig 1) for comparison with those estimated on supertree phylogenies (Fig 6). We use the tip-level pure-birth diversification rate (tip DR) metric [27] because it is readily calculable across all 10,000 trees in our credible sets while being highly correlated with model-based estimators of tip speciation rates (demonstrated in Quintero and Jetz [154] and reviewed in Title and Rabosky [28]). The reciprocal of the tip DR metric is a statistic called "equal splits" [24], which is tightly related to the "fair proportion" statistic commonly used to determine evolutionary distinctiveness (ED) (e.g., [39,40]). However, the ability to robustly estimate tip DR and ED requires trees that are completely sampled (contain all modern species) and probabilistically inferred (with uncertainty in topology and branch lengths).

Broadly, we find substantial heterogeneity in tip rates across the mammal tree, sometimes with a few high-tip-rate species nested among low-tip-rate species (Fig 1), resulting in long right-side tails in the tip rate distributions (positive skew, e.g., clades 38 and 44 in Fig 1). We find the consistently highest tip speciation rates in simian primates (clades 42–43 in Fig 1), including the human genus *Homo* (80th percentile, median 0.321 species/lineage/Ma; *Homo sapiens* and three extinct species) and Indo-Malayan lutung monkeys (95th percentile, 0.419, *Trachypithecus*). In contrast, species of *Ctenomys* tuco-tucos and *Pteropus* flying foxes display high tip speciation rates among otherwise slower-evolving species (clades 44 and 38, Fig 1). The evolutionarily distinctive platypus and aardvark have the lowest tip speciation rates (clades 1, 14; Fig 1). We suggest that tip rate skew is measuring aspects of within-clade speciation rate variation that may be otherwise uncaptured by model-fitting approaches (S5 Table). Future studies may thus find clade-level distributions of tip rates to be useful for comparative analysis.

Assessing how the different temporal frameworks of the node- and tip-dated backbones influence the species-level rate calculations (Fig 3B), we find that mammal species have broadly similar tip DR estimates across our tree sets. Indeed, there is approximately the same amount of variation in the 95% CIs of tip DR within a given tree set as between the two sets. The tip-dated phylogenies produce somewhat higher estimates of the tip DR harmonic mean (maximum of approximately 1.1 species/Ma versus approximately 0.8 in the ND phylogenies) but are nevertheless strongly correlated to the ND estimates (Spearman's $r$ = 0.93; linear model: $y = 0.02 + 1.17x$, $R^2$ = 0.85). The majority of the variation in tip rates among phylogenies appears to trace back to the younger node ages for mouse-related rodents and yangochiropteran bats (Fig 3A). Nevertheless, the internal consistency of each tree set suggests that applying either the tip- or ND phylogenies (or both) to a given comparative analysis would be appropriate.

We next compared the tip DR values from our backbone-and-patch analysis to estimates for the same species in previous supertrees to understand how different tree-building methods influence those rates (Fig 6; tree characteristics compared in Table 1). Overall, we find limited concordance between the tip rate estimates on our trees and the earlier supertrees of mammals (per-study tip rate correlations of $r$ = 0.60–0.62; Fig 6B). The 221 species identified in the top

**a** Species-level mammal trees

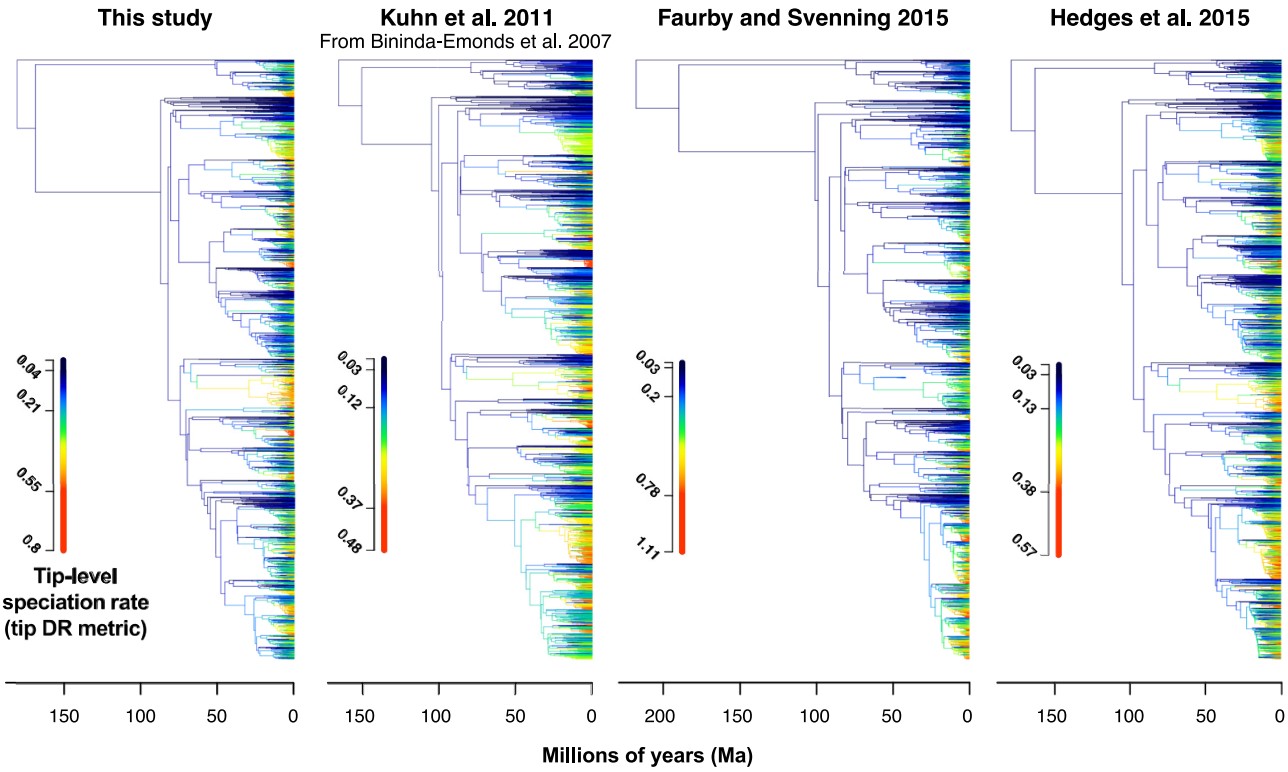

**b** Tip speciation rate comparison

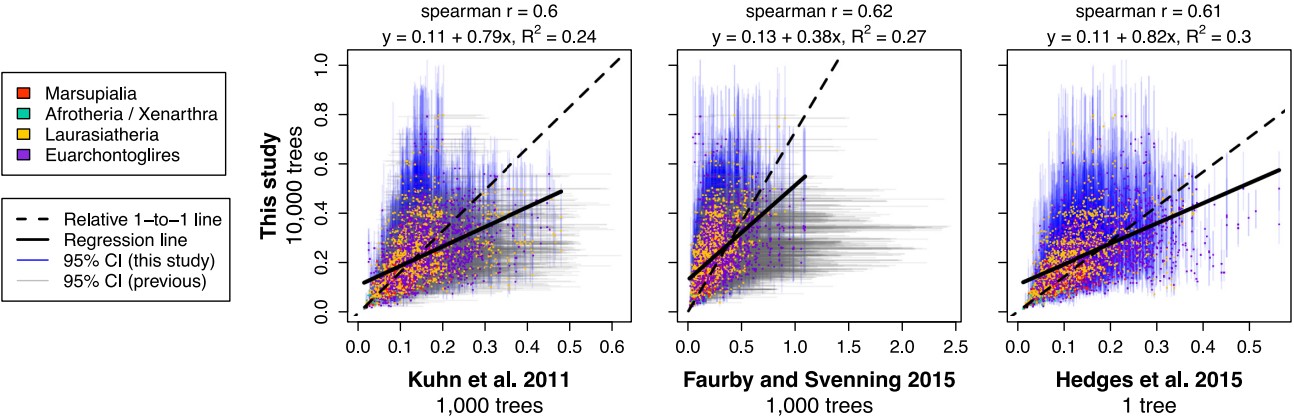

**Fig 6. Tip-level speciation rates (tip DR metric) from this study versus previous Mammalia-wide phylogenies.** Comparisons (a) as plotted on trees with relative color scales calibrated per data set so that the top 1% of the tip rate harmonic means correspond to the bright red color of each tree and (b) on a pairwise basis for all species with taxon names matching directly between data sets (*n* = 4,670, 5,329, and 5,033 species, respectively). Note that the x-axes differ but correspond to the range of tip DR values (95% CI) of each data set. Dryad data: https://doi.org/10.5061/dryad.tb03d03; phylogeny subsets: http://vertlife.org/phylosubsets. CI, confidence interval; tip DR, tip-level pure-birth diversification rate.

1% of tip DR values in our study are similarly recovered in the top percentile at a frequency of 12%, 21%, and 17% for the MRP supertree [87], DNA supertree [92], and consensus timetree [93], respectively. Reducing that comparison to the genus level, those 221 species belong to 46 different genera, of which 22%, 28%, and 41% are similarly recovered with at least one species

by those studies (same order). Thus, tip DR estimates from our study best match the DNA supertree at the species level and the consensus timetree at the genus level, although neither are close matches. Relative to each other, the MRP supertree and consensus timetree have the greatest similarity ($r = 0.78$ in pairwise rates versus $r = 0.57$–$0.59$ relative to the DNA supertree). Differences in tree estimation methodologies appear to drive differences in tip rates, although we acknowledge that differences in data availability at the time each study was conducted complicate our comparison.

Overall, the safest statement that we can make is that the tip rate estimates in our mammal trees are substantially different from those of previous trees. However, does that observation equate to our tip rate estimates of mammals being better? Value judgements are difficult in historical biology, in which we lack knowledge of the true evolutionary process. In the absence of simulation studies regarding the efficacy of the backbone-and-patch approach to tree building for recovering true rate dynamics (which would be a welcomed future contribution), we must rely on circumstantial arguments. We can couple the observation of tip DR differences between our trees and previous trees with the following pieces of evidence to make a determination: (1) temporal artifacts are incorporated as a result of supertree merging and polytomies (e.g., rodents Fig 5B); (2) there is a lack of rate uncertainty incorporated in supertrees, particularly when a credible set of trees is not generated (Figs 5 and 6); and (3) the present study was conducted with novel rigor regarding the gathering and cleaning of public DNA sequences, taxonomic reconciliation of synonymous names, supermatrix construction, and use of Bayesian inference methods at levels of the mammalian backbone and subclades (Fig 2).

Departing from past studies while improving the quality of data and inferences argues in favor of our Mammalia phylogenies being able to foster deeper insights into phylogeny-based questions in ecology, evolution, and conservation. Improved understanding of rate dynamics should enable causal hypotheses of biological diversification to be tested with greater reliability [155–157]. Tests that can exclude alternative hypotheses in the presence of realistic tree uncertainty should be viewed as providing durable knowledge regarding the historical trajectory of mammalian evolution.

## Limitations

This study was motivated by the clear need for mammalian phylogenetic hypotheses that contain comparably time-scaled branch lengths from root to tip. Until the computational challenges of inferring phylogeny from a genomic matrix of >6,000 species in a joint fossil-calibrated analysis can be overcome and more complete taxon sampling can be obtained, we suggest that the following sources of bias will limit our confidence in the resulting inferences.

**Missing data.**   The substantial level of missing data in our 31-gene supermatrix (mean = 88.1% per species) is worth further attention. Some simulation studies suggest that analyzing matrices with missing cells may yield erroneous estimates of topology, node support, and branch lengths (e.g., [158]), whereas other empirical and simulation studies have found no or small impact of missing data [159–162]. Wiens and Tiu [163] demonstrated that adding taxa with 90% missing data is beneficial to phylogenetic analyses when the alternative is to be misled by incomplete taxon sampling. Instead, model misspecification appears to have a greater impact on tree accuracy than missing data [162].

To empirically evaluate the impact of missing data, we performed a test of terminal branch length in the global ML tree relative to proportional DNA completeness (bp of sampled data per species / 39,099 bp of complete data). We found no relationship (Spearman's $r = -0.01$, $P = 0.582$), corroborating the result of Pyron and colleagues [161] that missing data do not consistently bias branch-length estimates. We note, however, that global biases in species

distributional knowledge (e.g., [164]) may additionally impact systematic attention and thus DNA completeness per taxon. Thus, future tests should aim to tease apart the relative impacts of missing data and their ecological covariates upon phylogenetic rate estimates.

**Tree completion and tip rates.** Tree "completion" methods are required to estimate tip rates if some modern species are unsampled for DNA. These methods include the simultaneous imputation of missing taxa during tree estimation (e.g., as we did in MrBayes patch clade analyses using Phylogenetic Assembly with Soft Taxonomic Inferences [PASTIS]-generated constraints [165]), as well as the use of per-clade sampling fractions to analytically integrate those missing species (e.g., as implemented in the Bayesian Analysis of Macroevolutionary Mixtures [BAMM] model [55]). Our approach using PASTIS is useful for obtaining taxonomically realistic tree shapes because branches for the DNA-missing species are drawn from the rate distribution informed by the local DNA matrix. We find up to 2×-higher variance in the tip DR estimates for the imputed species (S7 Fig, part a), which is an expected outcome because their placement in 10,000 trees is random within the specified taxonomic constraints. Tip DR medians for the same completed species are importantly no different than expected based on the range of tip DRs for DNA-sampled species (S7 Fig, part b). We thus find no bias in tip rates regarding whether a species was sampled for no genes or all 31 genes.

**Uneven taxonomic descriptions and tip rates.** Another possible bias in tip rate estimates is the disparate amounts of revisionary taxonomic attention that different clades of mammals have historically received. Taxonomic descriptions are arguably finer (i.e., more split) in larger- versus smaller-bodied mammals [166,167], but the many low–tip DR species among large and well-studied lemurs and carnivorans (Fig 1) suggests that taxonomy alone is not driving the apparent signal of fast, recent diversification in simian primates (clades 42 and 43; Fig 1). Many small mammals are being discovered, especially where biologists maintain active specimen-collection programs in the tropics (e.g., [168,169]), apparently without inflating rates. We include in our trees most of the 148 new species of Primates described in the last dozen or so years (28.6% of the extant total; [170]), which compares with 371 (14.5%), 304 (21.9%), and 86 (16.3%) new species of rodents, bats, and shrews, respectively, in that interval [170].

Importantly, we excluded most of the 227 new species of Artiodactyla described recently (41.1% of the total; [170]) because they nearly all derive from the monograph of Groves and Grubb [171] and are unvetted genetically [166,167,172]. We conservatively include 348 species in Artiodactlya rather than 551 [170,173] but still find elevated tip rates in whale- and cow-related lineages (clades 36 and 37, Fig 1), suggesting that those rates may be underestimates. Overall, we suspect that unequal taxonomic efforts should be less biasing in our mammal trees than in groups like amphibians (e.g., due to microendemism and greater tropical distributions; [174]), but future efforts to harmonize the definition of species-level lineages on a class-wide basis may nevertheless be fruitful.

## Recommended uses of the backbone-and-patch Mammalia trees

We recommend that researchers use the "completed" or "DNA-only" tree sets for addressing questions in which diversification rates or trait evolution are paramount, respectively; when that distinction overlaps (e.g., trait-dependent diversification), we recommend comparing analyses run on tree samples from both sets. In general, all types of analyses should be run on a sample of trees to meaningfully capture uncertainty. Even for questions of ancestral states or character evolution, it is still best to perform topology-based analyses on a sample of DNA-only trees rather than the consensus tree alone [33,34]. Rabosky [175] highlighted that in order to avoid biasing models of character evolution, the unsampled (DNA-missing) species should

be avoided because they will move around at random within genus- or family-level constraints [175,176]. Note, however, that species are generally sampled nonrandomly for DNA, so there is an alternative danger of excluding their trait values from analyses. Approaches that apply Rubin's rules to address missing data in traits and phylogenetic sampling are particularly promising, suggesting that sampling 50–100 trees is sufficient to meaningfully capture parameter uncertainty [177].

## Conclusion

The decade plus that has elapsed since the landmark publication of Bininda-Emonds and colleagues [81] highlights the clear need for improved approaches to species-level mammal phylogeny. Our novel, time-calibrated phylogeny incorporating all extant and described species of mammals now enables renewed focus on the causal factors underlying the historical tempo of evolutionary processes. However, the value of continued DNA sequencing for mammal species, as well as the further discovery and cladistic analysis of fossils, should not be understated. Continued improvements to the tree of life, and what we learn about mammalian biodiversity as a result, are directly dependent on the quality of input data. Inferring phylogenies that capture uncertainty in the reconstructed evolutionary process are essential to understanding our mammalian origins.

## Methods

We developed a 10-step strategy to build the Mammalia-wide tree sets (Fig 1; S1, S2, S3 and S4 Movies). As an overview (Fig 2A), we (1) sampled and vetted available DNA sequences for extant and recently extinct species, assembling them into a 31-gene supermatrix (steps 1–5); (2) developed an updated taxonomy accounting for 367 new species and 76 genus transfers (5,911 total species—Table 1, S2 Fig, and S1 Data); (3) built a global ML tree for 4,098 species with DNA to inform taxonomic constraints (step 6—S3 Data); (4) divided mammal diversity into 28 patch clades with nonoverlapping in-groups and identified lineages for use in the backbone (step 7—Table 4); (5) estimated patch clade phylogenies from DNA-only data sets and used taxonomic imputation to include 1,813 DNA-missing species (step 8—S4 Data); and (6) integrated fossil data at nodes and tips to compare methods of time-calibrating backbone divergences (step 9—S5 Data; Fig 2B). The full assembly of two sets of patch clades (DNA-only and completed) and two sets of backbones (ND and FBD; step 10) resulted in four sets of 10,000 trees for subsequent comparison of Mammalia-wide tree shape. In all four sets, the topological and age uncertainties in the backbone (Fig 2B–2E) are propagated to the 28 patch clades and full trees (see full data sets on Dryad: https://doi.org/10.5061/dryad.tb03d03).

### DNA-gathering pipeline

We used the Basic Local Alignment Search Tool (BLAST) algorithm [105] to query a local copy of NCBI's nucleotide database (downloaded on 20 April 2015), which allowed us to verify standards of homology and orthology among gathered sequence data. Use of BLAST to search for homologous genes avoided name-based searching by taxon or gene and the synonymy issues that entails [178]. We targeted 31 gene fragments commonly sampled among mammals (Table 2), using the family-level supermatrix of Meredith and colleagues [60] as our starting point (22 exons and five noncoding regions). Note that we treated RAG1 as two nonoverlapping regions (RAG1a and RAG1b) to match how researchers have most commonly published these sequences (e.g., GenBank accessions DQ865890 and AY011864 of *Didelphis virginiana*). To maximize species-level sampling, we targeted four protein-coding mitochondrial genes (mtDNA) in addition to the nuclear genes.

For each gene, we used a set of prevetted sequences, or "baits," as per-gene BLAST queries of the nucleotide database subset to the NCBI GI list for Mammalia (38,442,994 entries). Given that the per-gene baits were the basis for all downstream work, each bait set was carefully curated to ensure the valid homology of sequences to the targeted gene (e.g., robust alignment, absence of stop codons) while sampling up to one sequence per mammalian family. nDNA baits were taken from the mammal portion of Meredith and colleagues' [60] DNA alignment, as subset to one family representative per gene. We constructed our own mtDNA baits on a per-gene basis by identifying the longest vetted sequence per family from an output of general NCBI search terms [e.g., for ND2: "((((((txid40674[ORGN] ND2) NOT genome) NOT tRNA) NOT COI) NOT COIII) NOT cytb) NOT d-loop"]. For each gene, we used the "blastn" executable (BLAST+ v.2.2.31) with a coarse E-value of 10 when querying with DNA baits to ensure broad taxonomic coverage. The XML2 output format allowed us to assign NCBI taxonomic information [95] to each resulting hit for subsequent parsing. Using custom Bash scripts, we kept only the unique longest sequence per NCBI taxon ID that was greater than 200 bp in length. Parsing returned a sampling of ≥1 targeted genes for 6,247 unique taxon IDs at the ranks of species and subspecies out of a possible 7,319 such names (85%, as based on the NCBI taxonomy of 20 April 2015). Unaligned FASTA-format files for each gene were then subjected to a taxonomic matchup prior to alignment and further vetting (see S2 Table for steps of successive redundancy reduction, taxonomic matching, and error checking).

This initial procedure yielded direct matches for 4,725 of the 6,247 NCBI names (75%), of which we matched 765 via manual reference to the literature by consulting paper appendices in which the given sequences were published. These manual matches were species (304) or subspecies (410) or had ambiguous epithets (51; "cf.," "sp.," etc.). For 135 sequences (77 species), we manually matched accession numbers for which the corresponding NCBI taxon ID pertained to multiple valid species (denoted as "added manually" in S2 Table and S1 Data). Of the 5,490 NCBI names matched directly or manually, there were 1,273 junior synonyms, resulting in a starting list of 4,217 accepted species with ≥1 targeted gene sampled (3,954 species matching the International Union for the Conservation of Nature [IUCN] taxonomy + 263 added species; see below and S2 Fig). These DNA data were the basis for subsequent error checking.

### Taxonomy reconciliation and updating

Coherency among species names and their associated data is central to the integrity of any species-level comparative analysis. The NCBI taxonomy associated with our genetic data contained synonymous names and so needed to be vetted against an authoritative list of accepted mammalian species. We chose to initially base this matchup on the IUCN [94] because it (1) followed closely the authority of MSW3 [85], (2) was updated in several cases from MSW3, and (3) was tied to geospatial [179] and species trait [180] resources for downstream analysis. The IUCN base taxonomy contained 5,513 mammal species as downloaded 15 April 2015. We next used a synonym list compiled from several sources (Catalogue of Life [181], MSW3, IUCN; total of 195,562 unique equivalencies; updated from Meyer and colleagues [164]) to match the NCBI species and subspecies names to IUCN.

Our taxonomic matchup also revealed considerable changes to the number of valid mammal species because many were described after the approximately 2004 cutoff date of MSW3 [85] and approximately 2008 cutoff for most of the IUCN list. We made changes to the IUCN base taxonomy as follows: additions of (1) 367 new species, (2) 13 domestic species, and (3) 30 species recently extinct (last approximately 500 years) and subtraction of 12 species synonymized within existing IUCN names. The net change of 398 species resulted in a master

taxonomy of 5,911 mammalian species for this study, of which 5,804 species are considered extant (S1 Table and S2 Fig). The Mammal Diversity Database [170,173] (mammaldiversity. org) was an outgrowth of our project, yet it supersedes this total number of species because it continues to update mammalian taxonomy as new literature is published.

## Estimation of DNA-only and completed patch clades

Patch clades were estimated with (1) DNA sampling only and no topology constraints (4,098 species, "TopoFree") or (2) DNA sampling plus taxonomic constraints from the global ML tree to add the remaining unsampled species (5,911 species total, "TopoCons"). For both sets of patch clades, all phylogenies were estimated using identical models of evolution. Parameters of the GTR + G model were estimated independently among the nine partitions, as described above for the global ML tree. Although simpler partitioning strategies might have been selected for some (especially smaller) patch clades, we opted for consistency across all reconstructions. Molecular rate multipliers were estimated for each partition (ratepr = variable) to account for heterotachy (e.g., mtDNA second versus third positions [182,183]). We specified a relaxed clock model of each branch having an independent rate drawn from a gamma distribution [184] and length drawn from a birth–death process (brlenspr = clock:birthdeath). Ultrametric branch lengths were initially estimated in units of expected substitutions per site and subsequently rescaled to absolute time in millions of years from backbone divergence times (see below). Exponential distributions were given to priors for clock rate variance and net diversification (with mean of 0.1) and a beta prior (0–1) on the relative extinction rate. Note that previous backbone-and-patch studies fixed the relative extinction rate to zero, extinctionpr = fixed (0), which resulted in the patch clades being estimated under a pure-birth rather than birth–death process [27,39,40].

We set the in-group sampling probability to the proportion of sampled species per patch for the DNA-only analyses (Table 4) and to 1.0 for the taxonomically completed analyses. For each patch clade, we performed four parallel runs of MrBayes with BEAGLE, each run consisting of four chains of Markov chain Monte Carlo (MCMC; three heated and one cold) and sampled every 10,000 steps for 33,330,000 generations (S1 Text).

For the completed trees, taxonomic constraints for MrBayes were formed with the R package PASTIS [165]. This package reduced the potential for human error while also accounting for nonmonophyletic genera in the global ML tree. Ready-to-execute MrBayes files for each patch clade were generated from the following inputs (see Dryad data: https://doi.org/10.5061/dryad.tb03d03): (1) sequences file of aligned DNA in FASTA format, (2) taxa file of genus membership for all sampled and missing species, (3) missing clades file (if needed) designating where to constrain missing genera, (4) guide tree file giving the relationships of DNA-sampled species (global ML tree, pruned to patch clade species), and (5) template file specifying other MrBayes settings (e.g., rate priors, data partitions). Pruned portions of the global ML tree were not further altered, so all nodes present in that global tree—even those with low BS—were enforced as topology constraints for purposes of adding DNA-missing species to posterior phylogenies. This TopoCons procedure ensured that missing species were added in a phylogenetically informed way, although this has the trade-off of fixing the position of some poorly resolved nodes in the global ML tree. Topologies for those same nodes are sampled probabilistically in the TopoFree DNA-only patch clades.

From the input files, PASTIS adds DNA-missing species to the matrix block with "?" as the character datum for all aligned sites. If left unconstrained, those completed species would be placed at random throughout the posterior sample of trees. The flexible system of "hard," "negative," and "partial" constraints in MrBayes, when arranged hierarchically, was used to restrict

taxon additions to inside a given clade and outside other clades [50,165]. Partial constraints from PASTIS placed missing species randomly within the least inclusive clade containing all DNA-sampled members of a given genus. For example, the task of constraining 39 missing species of *Rattus* (of 66 total) was complicated by its paraphyly relative to *Diplothrix*, *Limnomys*, *Tarsomys*, *Bandicota*, and *Nesokia* (S3 Data). By using partial constraints, we could constrain missing *Rattus* species to their paraphyletic grouping across the posterior distribution. Completed species' branch lengths were drawn from the same birth–death distribution as the rest of the patch clade, biasing PASTIS completions toward rate-constant processes while preserving the taxonomically expected tree shape [27,165]. Therefore, our phylogenies should tend to favor null explanations involving constant-rate species diversification.

### Fossil-dated backbone trees

Divergence times and evolutionary relationships among basal lineages of mammals served as the "backbone phylogeny" to which all patch clades were rescaled to absolute time and then joined to form full species-level trees of Mammalia (Fig 2). Fossil information provided the temporal framework for calibrating backbone divergences. Two types of backbones were constructed: (1) ND, using 17 fossil calibrations and one root constraint from Benton and colleagues [139], as augmented by Philips ([185]; S1 Text, section 6 for list of fossil calibrations), and (2) tip-dated (FBD [58]), using the morphological data set of Zhou and colleagues [119] trimmed to 76 fossil (mostly Mesozoic fossils, 66–252 Ma) and 22 extant taxa. In both types of analyses, we focused on a common set of extant taxa to subset the full supermatrix for molecular characters (59 mammals, representing each of the 28 patch clades plus select additional family-level taxa, and one out-group, *Anolis carolinensis*). Taxa were selected based on their extent of genetic sampling, which for most taxa was >25 of 31 genes (median: 29, range: 3–31). Some additional taxa were selected so that nodes were present for subsequent age constraint in the node-dating analyses or due to their inclusion in the morphological data set of Zhou and colleagues [119].

Node age priors were based on the "best practices for fossil calibrations" recommendations of Parham and colleagues [57], which among other criteria states that (1) fossils should be confidently placed in the crown group of the calibrated node using a formal cladistic analysis of extant and fossil morphological characters and that (2) fossils placed along the stem of a given crown group can only inform the minimum age of the next node back (sister of crown groups [186]). Calibrations were set either as (1) exponential priors ("NDexp"), offset to minimum ages with soft maxima [187] so that the upper 95% of the distribution equaled maximum ages [formula based on the exponential distribution: exp mean = $(-\ln[0.05]/[\text{max-min}])^{-1}$], or (2) as uniform priors ("NDuni") spanning minima to maxima. These strategies of exponential versus uniform priors were compared to test the sensitivity of dating results. Similar node ages led us to focus on the more conservative NDexp backbone for comparison to the tip-dated backbone and downstream diversification-rate analyses. Analyses were run in MrBayes v.3.2.6 similarly as the patch clades but with the following exceptions: (1) the taxon sampling probability was set to 0.0102 (59 of 5,804 extant species), (2) the sampling strategy was set to "diversity" given our maximization of taxonomic diversity in the backbone phylogeny, and (3) the birth–death clock rate prior was set to a lognormal that assumed each nucleotide site changed one time in the approximately 318 million–year root-to-tip distance (mean = log[1/318], standard deviation = exp[1/318]) [188]. The node age prior was set to "calibrated," indicating that the probability distribution on terminal and interior node ages was derived from the calibration settings. Note that the birth–death process with variable rates per molecular partition was also implemented here, the same as with the patch clades. We conducted four independent

runs of four chains each (three heated and one cold), run for 50,000,000 generations and sampled every 10,000 generations.

For tip dating, we aimed to replicate the study of Zhou and colleagues [119] with the additional "total evidence" perspective of our molecular data set. Tip-dating methods reconstruct the phylogeny of living and fossil taxa according to morphological characters coded for both taxon classes, using the stratigraphic ranges of fossils to inform the birth–death clock model [123,189]. Fossilization can now be parameterized along with birth and death (FBD; [58]), and diversified sampling of extant and fossil taxa can be accommodated in the FBD process [190]. A key contrast in tip dating versus node dating is that fossils impact node ages via their cladistic placement. Thus, different (in some cases fewer) assumptions are involved in tip than in node dating. However, because the former methods are in greater flux [64,121,191], comparing both strategies is useful. Analyses were run in MrBayes v.3.2.6 in a manner analogous to node dating but with the following exceptions: (1) each fossil tip was given a uniform calibration prior between minimum and maximum stratigraphic ages; (2) the "clock:fossilization" branch-length prior was specified, as appropriate for clock trees including fossils [58,190]; and (3) no node calibrations were enforced. Whereas node-dating analyses required topology constraints for each calibration point, none were required in FBD. However, to be sure that our backbone topology matched that recovered in Zhou and colleagues [119], we used hard constraints on the following nodes: in-group from *Anolis* out-group; crowns (extant taxa) and total groups (with stem fossils) for Placentalia, Marsupialia, and Monotremata; and constraints on the crowns of Theria and Mammalia to ensure the placement of haramiyidans in Mammaliaformes but outside crown mammals ([118,119]; contra to [192]). Note that by exactly following the topology of Zhou and colleagues [119], we constrained two shuotheriids (*Shuotherium* and *Pseudotribos*) to be paraphyletic because of an apparent error in their matrix [120]. As with the node-dating runs, for tip dating we conducted four independent runs of four chains each (three heated and one cold), each run for 50,000,000 generations and sampled every 10,000 trees (executables in S5 Data). We estimate that the final runs of node- and tip-dating backbone analyses (not counting troubleshooting) together took 8 weeks in MrBayes using 16 BEAGLE-enabled nodes for a total of approximately 21,500 cpu hours.

## Construction of full dated mammalian phylogenies

We summarized the Bayesian patch- and backbone-level runs after discarding the first 25% and 50% of samples as burn-in, respectively. Analyses demonstrated convergent traces in Tracer v.1.5 potential scale reduction factors (PSRFs) of approximately 1 among chains and ESS scores >200 for most parameters, allowing us to combine the four independent runs. Each run was reduced to 2,500 sampled trees after burn-in (from 3,333 for patch clades and 5,000 for backbones), yielding posterior distributions of exactly 10,000 trees upon combining in the Bash shell. For the backbone analyses, fossil-calibrated branch lengths were finalized to absolute time units (millions of years) using the "burntrees.pl" script "--myr" flag (https://github.com/nylander/Burntrees).

To unite the 28 patch clades and the backbone phylogeny (Fig 2A), we first needed to rescale the patch clade distributions of 10,000 trees to absolute time in millions of years. Because patches were estimated in ultrametric but relative units—brlenspr = clock:birthdeath, clockratepr = Fixed(1.0)—this procedure was accomplished with a simple multiplication in R with the "ape" package [193]. We used the rescale-and-graft procedure of previous studies [27,39,188], outlined here:

1. Load all 10,000 trees for a selected backbone analysis and each of the patch clades.

2. Prune all backbones to *A. carolinensis* and 28 species representatives, one for each of the patch clades.

3. Prune the placeholder taxa from Dermoptera and Platacanthomyidae patch clades.

4. For each of the 10,000 samples of backbones and patches (sequentially),

   a. take one pruned backbone tree; get branching times and identify pairwise MRCA nodes that correspond to the out-group-to-in-group relationship, recording the node age per patch (this root time is used to rescale the patch clade to Ma);

   b. take 28 trees, one from each patch clade; for each, get the relative branching times and divide the age of the in-group crown by that of the root, obtaining the relative scale of the stem edge from out-group to in-group; prune outgroups;

   c. for each patch clade, multiply root time (in Ma) from the backbone by the relative scale of the patch clade to obtain the absolute scale of the stem edge leading to the patch crown; divide all edge lengths by the maximum node height to obtain relative edge lengths of the patch clade and then multiply those by the absolute scale of the stem to rescale all patch clade branches to time in Ma;

   d. use which.edge() in ape to identify the tip edge in the pruned backbone upon which each patch clade is to be grafted; shorten those branches by subtracting the absolute scale of the stem edge per clade; and

   e. use bind.tree() in ape to graft each rescaled patch clade to the corresponding shortened edge of the pruned backbone, thereby forming one uniformly time-scaled tree of 5,911 (or 4,098) species of mammals.

5. Repeat that procedure to construct credible sets of 10,000 phylogenies for each of the backbone analyses (ND and FBD). Note that some authors call these "pseudoposterior" trees because the final trees graft together independent inferences.

Exceptions to the above procedure involved Monotremata, Marsupialia, and Lagomorpha, in which patch clades were rescaled to crown rather than stem divergences because of the use of a distant out-group (*Rattus*), and some of the bat and rodent patch clades were conditionally assembled because of basal phylogenetic uncertainty (see S1 Text, section 7).

For purposes of visually summarizing the variation in the credible sets of full trees, we (1) constructed MCC consensus trees in TreeAnnotator v.1.8.2 [194] and (2) generated movies to cycle through the variation in tree topology and node ages for trees in each credible set. Mean node ages and 95% HPD intervals were summarized for the backbone-level MCC trees for use in node age comparisons with other studies. For the full species-level trees, we kept the target node heights of the MCC tree rather than annotating node averages (in some cases, node averages would collectively result in negative branch lengths). Although the species-level DNA-only MCC trees are appropriate for some analyses, analyzing a credible set of trees is generally still preferred. The completed MCC trees were for display purposes only (e.g., Fig 1) because the DNA-missing species have variable placements within taxonomic constraints, and thus, no single tree can meaningfully summarize their uncertainty (see Discussion section on "Recommended uses"). Movies of the credible trees' sets were generated as animated gifs from png-formatted plots of 100 sampled trees using the R package "magick" [195].

## Tip-level speciation rates

To characterize species-level variation in speciation rates across Mammalia and compare it among studies, we calculated per-species estimates of expected pure-birth diversification rates for the instantaneous present moment (tips of the tree) using the inverse of the equal splits measure [27,196]. This metric has been called the "DR statistic" and "tip-level diversification rate" because it measures recent diversification processes among extant species [154]. However, to avoid confusion with "net diversification," for which tip DR is misleading when extinction is very high (relative extinction >0.8 [197]), we here refer to tip DR as a tip-level speciation rate estimator. Tip DR emphasizes geologically recent speciation over deeper-time dynamics, and so it is comparatively less prone to bias from undetected extinction events or nonidentifiability [198] than methods for detecting branch-specific or tree-wide rate shifts [199–201]. We calculate tip DR on full Mammalia phylogenies from the root to each tip as

$$Tip\ DR_i = 1/\sum_{j=1}^{N_i} l_j \frac{1}{2^{j-1}}$$

where $N_i$ is the number of edges on the path from tip $i$ to the root, and $l_j$ is the length of edge $j$. This equation assumes a fully bifurcating tree [24,27]. Because $j = 1$ is the pendant edge leading to tip $i$, that branch length carries the greatest weight on the resulting value, with every ensuing rootward edge discounted exponentially as it is shared with other species. Sister species thus have identical tip DR values. Species with the highest tip DR have many short branches shared with other species near the present, implying that recent branching is abundant, whereas low–tip DR species are subtended by long unshared branches (i.e., they are evolutionarily distinct [24]).

We compared our estimates of tip DR calculated across 10,000 trees to those derived from the MRP supertree (1,000 trees; [87]), DNA supertree (1,000 trees; [92]), and consensus time-tree of mammals (1 tree; [93]). We summarized the harmonic mean and 95% confidence interval (CI) for the tip DR value of each species in each data set for subsequent plotting. Note that tip DR should only be calculated on "completed" trees that account for all extant and recently extinct species, so we excluded 245 species of Pleistocene extinct mammals in the Faurby and Svenning [92] supertree, retaining only those 97 species presumed to have gone extinct in the last approximately 500 years. Direct matching of species binomial names from our study ($n$ = 5,911) to the other data sets yielded 4,670, 5,329, and 5,033 pairwise comparisons of tip DR, respectively, for these three studies listed above.

## Lineage accumulation comparisons

To visualize variation in branching times across Mammalia and among studies, we randomly selected 100 trees from each study and plotted the accumulation of lineages through time from root to tip. We used the ltt.plot() and ltt.lines() functions in the R package ape to output png files for publication.

To assess the congruence of our molecular phylogeny–based rate estimates with the fossil record, we analyzed Mammalia fossil occurrence data from the Paleobiology Database [132], as downloaded on 16 August 2018. Grouping by genus after excluding ichnotaxa and uncertain genera, we recovered 71,928 occurrences of 5,300 genera from the earliest basal Mammaliaformes (e.g., *Gondwanadon*, late Triassic approximately 235 Ma) to modern genera with fossil records (e.g., *Pteropus*). The maximum stratigraphic age of the oldest genus per extant mammalian order was used to represent the "fossil stem maximum" age for comparison with our estimates and those of previous studies.

## Data availability

All curated data and code are available in the supplementary materials deposited in the Dryad Digital Repository: https://doi.org/10.5061/dryad.tb03d03 [202]. Code for reproducing analyses and figures is also on Github: https://github.com/n8upham/MamPhy_v1. Credible sets of 10,000 trees are available for taxonomic subsetting using the online tool at https://vertlife.org/phylosubsets, and further mammal tree visualizations are presented at http://vertlife.org/data/mammals/.

## Supporting information

**S1 MDAR Checklist. Included are details of the data availability for this study.** MDAR, Materials Design Analysis Reporting.
(PDF)

**S1 Fig. Trade-off between tree size (number of analyzed tips) and the amount of statistical uncertainty in the tree.** Computational costs increase with tree size and the realism of the evolutionary models, resulting in reduced ability to propagate estimate uncertainty in larger trees. We suggest a current upper limit of approximately 1,000 species for Bayesian coestimation, beyond which performing supermatrix analyses requires our backbone-and-patch approach (green) to divide the tree into smaller subanalyses.
(TIFF)

**S2 Fig. Our master taxonomy (5,911 species) results from uniting the mammal lists of NCBI (genetic data) and IUCN (name authority).** Bars representing each taxonomic list are sized proportionately to the number of names in each list category. We conducted a baited BLAST search of NCBI using 31 genes, from which data was returned for approximately 85% of the NCBI names (6,247 of 7,319; list of 20 April 2015). Steps of synonym matching and taxon addition to the IUCN list resulted in our master taxonomy. Black corresponds to the 4,217 species initially with DNA, including 410 species we added to the IUCN list (newly described species, domestic forms, or recently extinct), which was reduced to 4,098 species after error-checking steps (see S1 and S2 Data; Dryad: https://doi.org/10.5061/dryad.tb03d03). IUCN, International Union for the Conservation of Nature; NCBI, National Center for Biotechnology Information.
(TIFF)

**S3 Fig. Final genetic sampling of mammal species from NCBI relative to total diversity of genera and families.** Species were considered sampled for DNA if one or more of our 31 genes were sampled in the final supermatrix after DNA cleaning, error checking, and taxonomic reconciliation. Inset are detailed views of the genetic sampling within the 20 most speciose genera (of 1,283 total) and families (of 127 total). Much additional DNA sequencing is needed to move from the 4,098 species with genetic sampling to the >6,000 species of modern mammals. Dryad data: https://doi.org/10.5061/dryad.tb03d03. NCBI, National Center for Biotechnology Information.
(TIFF)

**S4 Fig. Full node-dated backbone phylogenies.** These were constructed using (a) exponential node priors (NDexp) or (b) uniform priors (NDuni) in MrBayes based on 17 fossil calibrations and molecular data from our 31-gene supermatrix. Topology is the maximum clade credibility tree of 10,000 phylogenies. Median ages and 95% highest posterior density intervals are displayed at nodes. Node circles indicate PP values of ≥0.95 (black), 0.94–0.75 (gray), and <0.75 (white). Dryad data: https://doi.org/10.5061/dryad.tb03d03. PP, posterior probability.
(TIFF)

**S5 Fig. Full tip-dated backbone phylogeny.** This was constructed using FBD in MrBayes based on the morphological matrix of Zhou and colleagues [119] and molecular data from our 31-gene supermatrix. Topology is the maximum clade credibility tree of 10,000 phylogenies. Median ages and 95% highest posterior density intervals are displayed at nodes. Node circles indicate PP values of ≥0.95 (black), 0.94–0.75 (gray), and <0.75 (white). Dryad data: https://doi.org/10.5061/dryad.tb03d03. FBD, fossilized birth–death; PP, posterior probability. (TIFF)

**S6 Fig. Comparison of results from three methods used to time-calibrate the backbone.** Each method is pruned to the 28 patch clade representatives: (a) FBD where fossil taxa are placed as extinct tips in the tree (left side) and then pruned (right side); and ND approaches setting priors as (b) exponential priors from minimum to soft-max ages and (c) uniform priors spanning minimum to maximum ages. Trees are maximum clade credibility summaries of 10,000 trees. Circles at nodes indicate PP values of ≥0.95 (black), 0.94–0.75 (gray), and <0.75 (white), with the values < 0.95 given. (d) Inferred ages for backbone nodes are compared across methods, as based on the ND tree. Note that the FBD trees did not recover node 55 (see part a and S5 Fig). Dryad data: https://doi.org/10.5061/dryad.tb03d03. FBD, fossilized birth–death; ND, node-dated; PP, posterior probability. (TIFF)

**S7 Fig. Effect of gene sampling per species upon tip DR estimates.** Compared are the per-species (a) variances and (b) medians in tip DR across 10,000 node-dated trees versus the number of genes (0–31) sampled in the global DNA supermatrix. Completed trees are those in which no-DNA species (0 genes) were added using PASTIS during MrBayes runs. As expected, variance in tip DR estimates is higher for completed species (note the different y-axes from left to right panel in part a). However, median tip DR estimates are similar between completed and DNA-only trees. Spearman's correlation coefficients, $r$, are shown for each plot as an indication of general trends in the data (slight negative trends do not account for phylogenetic covariance). Dryad data: https://doi.org/10.5061/dryad.tb03d03. tip DR, tip-level pure-birth diversification rate; PASTIS, Phylogenetic Assembly with Soft Taxonomic Inferences. (TIFF)

**S8 Fig. Summary of data contributions per author on the NCBI public sequence database.** The top 30 contributors as first and last authors toward the 31-gene supermatrix used in this study, first as barplots of author frequency (top row) and then treemap diagrams of the same data (bottom row). The full DNA supermatrix consisted of 21,021 total sequences after error-checking steps, of which 1,963 sequences were contributed by the Meredith and colleagues [60] study that served as DNA baits for 27 of the 31 genes examined in this study. Dryad data: https://doi.org/10.5061/dryad.tb03d03. NCBI, National Center for Biotechnology Information. (TIFF)

**S9 Fig. Consensus tree of the DNA-only node-dated phylogeny of 4,098 mammal species.** This MCC tree summarizes the mean node ages, 95% highest posterior densities, and nodal support values across 10,000 node-dated trees. Dryad data: https://doi.org/10.5061/dryad.tb03d03; phylogeny subsets: http://vertlife.org/phylosubsets. MCC, maximum clade credibility. (EPS)

**S10 Fig. Consensus tree of the DNA-only tip-dated phylogeny of 4,098 mammal species plus 76 stem fossils.** This MCC tree summarizes mean node ages, 95% highest posterior

densities, and nodal support values across 10,000 tip-dated trees. Dryad data: https://doi.org/10.5061/dryad.tb03d03; phylogeny subsets: http://vertlife.org/phylosubsets. MCC, maximum clade credibility.
(EPS)

**S1 Movie. Visual summary of phylogenetic uncertainty in the node-dated completed trees.** Variation in the tree topology and node ages is shown across 100 trees sampled from the credible set of 10,000 trees, including the relative placement of higher taxa on the Mammalia backbone (colored balls) and the location of taxonomically imputed species (gray bars on tips). Dryad data: https://doi.org/10.5061/dryad.tb03d03; phylogeny subsets: http://vertlife.org/phylosubsets; direct link to mammal tree visualizations: http://vertlife.org/data/mammals/.
(MOV)

**S2 Movie. Visual summary of phylogenetic uncertainty in the tip-dated completed trees.** Variation in the tree topology and node ages is shown across 100 trees sampled from the credible set of 10,000 trees, including the relative placement of higher taxa on the Mammalia backbone (colored balls) and the location of taxonomically imputed species (gray bars on tips). Dryad data: https://doi.org/10.5061/dryad.tb03d03; phylogeny subsets: http://vertlife.org/phylosubsets; direct link to mammal tree visualizations: http://vertlife.org/data/mammals/.
(MOV)

**S3 Movie. Visual summary of phylogenetic uncertainty in the node-dated DNA-only trees.** Variation in the tree topology and node ages is shown across 100 trees sampled from the credible set of 10,000 trees, including the relative placement of higher taxa on the Mammalia backbone (colored balls; all modern species are sampled for DNA). Dryad data: https://doi.org/10.5061/dryad.tb03d03; phylogeny subsets: http://vertlife.org/phylosubsets; direct link to mammal tree visualizations: http://vertlife.org/data/mammals/.
(MOV)

**S4 Movie. Visual summary of phylogenetic uncertainty in the tip-dated DNA-only trees.** Variation in the tree topology and node ages is shown across 100 trees sampled from the credible set of 10,000 trees, including the relative placement of higher taxa on the Mammalia backbone (colored balls; all modern species are sampled for DNA). Dryad data: https://doi.org/10.5061/dryad.tb03d03; phylogeny subsets: http://vertlife.org/phylosubsets; direct link to mammal tree visualizations: http://vertlife.org/data/mammals/.
(MOV)

**S1 Table. The master taxonomy of this study versus existing authoritative lists.** Common authorities for mammals are MSW3 [85], IUCN [94], and the MDD [170]. IUCN, International Union for the Conservation of Nature; MDD, Mammal Diversity Database; MSW3, *Mammal Species of the World*, *third edition*.
(DOCX)

**S2 Table. Results from BLAST searches for each of the 31 gene fragments used in this study.** Successive steps to parse results to unique NCBI species and subspecies names, match NCBI names to initially accepted names in the master taxonomy, and then manual addition (+) and removal (−) steps of error checks to yield per-gene final accepted species. excl., excluded; NCBI, National Center for Biotechnology Information.
(DOCX)

**S3 Table. Divergence times relative to prior studies.** Crown divergence mean Est and 95% CI (lower and upper) for each taxon listed, with the 27 extant mammal orders in capital letters.

Ages in gray are order-level divergences estimated near the K-Pg extinction, with "near" defined as having 95% CI < 3 Ma of 66 Ma, whereas black ages have CIs >3 before 66 Ma. Our node-dated estimates are compared with global amino acid and DNA dates [60], best estimate dates [81], combined "14K+Mit" dates [61], rapid diversification posteriors [64], and fossil compendia [63,132]. Dates are missing if a node was not recovered or lacked taxon sampling. CI, confidence interval; Est, estimate; K-Pg, Cretaceous–Paleogene; Ma, million years ago.
(DOCX)

**S4 Table. Fossil maximum stratigraphic ages per order relative to stem ages from our node-dated phylogeny (95% HPD age of 10,000 trees).** Fossil occs. per extant mammalian order were gathered from the Paleobiology Database. HPD, highest posterior density; occ., occurrences.
(DOCX)

**S5 Table. Per-clade summary of tip DR.** Tip DR median, 95% confidence interval, and the skew in a given clade across 10,000 node-dated trees. Tests of the clade tip DR versus the (non-clade) background rate used the Mann–Whitney $U$ statistic: greater (>, grayed), lesser (<), or NS. tip DR, tip-level pure-birth diversification rate; NS, not significant.
(DOCX)

**S1 Data. Details of the DNA cleaning steps and updated master taxonomy of mammals used in this study.** Three multitab Excel files, including the per-gene sampling in the final supermatrix, initial gene lengths, NCBI accession numbers, and the authors of each sequence. Dryad data: https://doi.org/10.5061/dryad.tb03d03. NCBI, National Center for Biotechnology Information.
(ZIP)

**S2 Data. Per-gene DNA alignments for 31 genes and gene tree outputs from RAxML.** Includes PDF plots of each gene tree along with the newick tree file and the DNA alignments in phylip format. Dryad data: https://doi.org/10.5061/dryad.tb03d03. RAxML, Randomized Axelerated Maximum Likelihood.
(ZIP)

**S3 Data. Global ML tree for 4,098 species of mammals built from the 31-gene supermatrix.** Includes the full 31-gene supermatrix alignment, taxonomy file, newick tree file and PDF plotting of global RAxML tree, and R code for dividing the tree into patch clade segments for scaffolding the subsequent Bayesian analyses. Dryad data: https://doi.org/10.5061/dryad.tb03d03. ML, maximum-likelihood; RAxML, Randomized Axelerated Maximum Likelihood.
(ZIP)

**S4 Data. Results of 28 patch clade phylogenies in relative time across Mammalia.** Relative time results (i.e., not yet time scaled) presented as MCC trees for each of the patch clade runs, both as nexus and PDF plots, and details of the species and gene sampling in each patch clade. Dryad data: https://doi.org/10.5061/dryad.tb03d03. MCC, maximum clade credibility.
(ZIP)

**S5 Data. Results and run files for backbone divergence-time analyses in MrBayes.** Time-scaled MCC trees for each of the three backbone dating analyses in nexus format, as well as the run files and details of taxon sampling and node constraints. Dryad data: https://doi.org/10.5061/dryad.tb03d03. MCC, maximum clade credibility.
(ZIP)

**S1 Text. Supplementary methods and results for the analyses conducted in this study.**
Included are details of the taxonomic matchup, DNA sequence alignment, gene tree construction and error checking, final DNA sampling, 31-gene supermatrix analyses, patch clade run settings, 17 node-dated fossil calibrations, and construction of the full Mammalia-wide phylogenies.
(PDF)

## Acknowledgments

We thank I. Quintero, M. Landis, A. Mooers, A. Pyron, G. Thomas, D. Greenberg, and E. Florsheim for conceptual discussions that improved this study; B. Patterson, D. Schluter, K. Rowe, J. Brown, T. Colston, T. Peterson, D. Field, T. Stewart, and J. Davies for comments on earlier drafts; S. Upham for improving figure design; C. Meyer for his synonym list; and M. Koo, A. Ranipeta, J. Hart, M. Swanson, C. Burgin, and J. Colella for database assistance. We additionally thank the 6,069 individual authors of 1,934 published studies on Genbank—and the numerous natural history museums housing those mammal specimens—for contributing the genetic data that enabled this synthetic study.

## Author Contributions

**Conceptualization:** Nathan S. Upham.

**Data curation:** Nathan S. Upham, Jacob A. Esselstyn.

**Formal analysis:** Nathan S. Upham.

**Funding acquisition:** Jacob A. Esselstyn, Walter Jetz.

**Investigation:** Nathan S. Upham.

**Methodology:** Nathan S. Upham.

**Project administration:** Walter Jetz.

**Supervision:** Walter Jetz.

**Validation:** Nathan S. Upham, Jacob A. Esselstyn.

**Visualization:** Nathan S. Upham.

**Writing – original draft:** Nathan S. Upham.

**Writing – review & editing:** Nathan S. Upham, Jacob A. Esselstyn, Walter Jetz.

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
