## [Editor Report · Decision Letter 0]

29 Jul 2019

Dear Dr Upham, 

Thank you for submitting this revised version of your manuscript entitled "Inferring the mammal tree: species-level sets of phylogenies for questions in ecology, evolution, and conservation" for consideration as a Methods and Resources by PLOS Biology.

I am writing on behalf of my colleague, Senior Editor Dr Roland Roberts who is currently on vacation. He did assess your submission last week though and consulted with the original Academic Editor, and I am writing to let you know that we would like to send your submission out for external peer review, we will try to sign back up the previous reviewers for your manuscript.

*Please be aware that, due to the voluntary nature of our reviewers and academic editors, manuscripts may be subject to delays during the holiday season. Thank you for your patience.*

**Important**: Please also see below for further information regarding completing the MDAR reporting checklist. The checklist can be accessed here: https://plos.io/MDARChecklist

Please re-submit your manuscript and the checklist, within two working days, i.e. by Jul 31 2019 11:59PM.

Kind regards,

Emma Ganley PhD

Chief Editor, PLOS Biology

eganley@plos.org

On behalf of:

Senior Editor

PLOS Biology

INFORMATION REGARDING THE REPORTING CHECKLIST:

PLOS Biology is pleased to support the "minimum reporting standards in the life sciences" initiative (https://osf.io/preprints/metaarxiv/9sm4x/). This effort brings together a number of leading journals and reproducibility experts to develop minimum expectations for reporting information about Materials (including data and code), Design, Analysis and Reporting (MDAR) in published papers. We believe broad alignment on these standards will be to the benefit of authors, reviewers, journals and the wider research community and will help drive better practise in publishing reproducible research. 

We are therefore participating in a community pilot involving a small number of life science journals to test the MDAR checklist. The checklist is intended to help authors, reviewers and editors adopt and implement the minimum reporting framework. 

IMPORTANT: We have chosen your manuscript to participate in this trial. The relevant documents can be located here:

MDAR reporting checklist (to be filled in by you): https://plos.io/MDARChecklist

**We strongly encourage you to complete the MDAR reporting checklist and return it to us with your full submission, as described above. We would also be very grateful if you could complete this author survey:

https://forms.gle/seEgCrDtM6GLKFGQA

Additional background information:

Interpreting the MDAR Framework: https://plos.io/MDARFramework

Please note that your completed checklist and survey will be shared with the minimum reporting standards working group. However, the working group will not be provided with access to the manuscript or any other confidential information including author identities, manuscript titles or abstracts. Feedback from this process will be used to consider next steps, which might include revisions to the content of the checklist. Data and materials from this initial trial will be publicly shared in September 2019. Data will only be provided in aggregate form and will not be parsed by individual article or by journal, so as to respect the confidentiality of responses. 

Please treat the checklist and elaboration as confidential as public release is planned for September 2019.

We would be grateful for any feedback you may have.

---

## [Decision Letter · Decision Letter 1]

6 Sep 2019

Dear Dr Upham,

Thank you for submitting your revised Methods and Resources entitled "Inferring the mammal tree: species-level sets of phylogenies for questions in ecology, evolution, and conservation" for publication in PLOS Biology. I have now obtained advice from two of the original reviewers and have discussed their comments with the Academic Editor. 

Based on the reviews, we will probably accept this manuscript for publication, assuming that you will modify the manuscript to address the remaining concerns raised by the reviewers (and the additional guidance from the Academic Editor at the foot of this letter).

IMPORTANT: As well as addressing the reviewers' remaining concerns, please could you ensure that the URLs for your Dryad and Github depositions are clearly cited as the site of the underlying data and code in each of the relevant Figure Legends?

We expect to receive your revised manuscript within two weeks. Your revisions should address the specific points made by each reviewer. In addition to the remaining revisions and before we will be able to formally accept your manuscript and consider it "in press", we also need to ensure that your article conforms to our guidelines. A member of our team will be in touch shortly with a set of requests. As we can't proceed until these requirements are met, your swift response will help prevent delays to publication.

Please note that you may have the opportunity to make the peer review history publicly available. The record will include editor decision letters (with reviews) and your responses to reviewer comments. If eligible, we will contact you to opt in or out.

Early Version: Please note that an uncorrected proof of your manuscript will be published online ahead of the final version, unless you opted out when submitting your manuscript. If, for any reason, you do not want an earlier version of your manuscript published online, uncheck the box. Should you, your institution's press office or the journal office choose to press release your paper, you will automatically be opted out of early publication. We ask that you notify us as soon as possible if you or your institution is planning to press release the article.

Sincerely,

Roli Roberts

Senior Editor

PLOS Biology

REVIEWERS' COMMENTS:

Reviewer #2:

[identifies himself as Olaf R.P. Bininda-Emonds]

With the revision, the MS has been sharply refocussed on the topology of the mammalian timetree with the macroevolutionary analyses being reduced to discussions of diversification rates. As such, there is much more room for a discussion about how the timetree was generated and how it compares to other mammalian timetrees of similar scale, which I welcome greatly. Much of the information in this regard in the previous version was very limited because of space limitations and I do find this topic to be particularly important, especially with the authors contending previously that their tree was a large improvement over other efforts. The results here instead show much more agreement between these trees and much more uncertainty within the new timetree, which seems to be a accurate reflection of the state of affairs.

This is not to say that I do not find the new timetree to be impressive. I certainly do! I did not expect there to be a DNA-based phylogeny of mammals on such a scale for the next 10 years perhaps. so this study, at least from my perspective, is way ahead of its time!

In general, I am very satisfied with the revisions and the additional information that the authors could provide about the timetree and have one major comment and several more minor ones.

The major comment

Although I think that this tree is a great advancement over previous supertrees, I also feel that the authors are being unfair in their criticism of supertrees. Contrary to what the authors repeatedly state, polytomies are not inherent to supertrees, but rather to the entire non-Bayesian framework in which the suer trees were conducted. Even supermatrices computed with MP and theoretically with ML will contain polytomies because a non-Bayesian framework ignores equally optimal trees in general to focus on the consensus solution. More importantly, it doesn't have to. If one were to look at all equally most optimal solutions or to use the set of bootstrap trees that were generated, one would get a rough equivalent to the credible sets of trees of a Bayesian analysis. "Ignoring" or "hiding" uncertainty is not inherent to supertrees or other non-Bayesian analyses, but rather a choice that is probably rooted partly historically when computer performance was insufficient to handle subsequent analyses on so many trees. That other investigators misinterpret what a soft polytomy means (PG 29) is not the fault of the investigators who quite clearly stated that these were soft polytomies and knew what this meant.

The minor comments

PG 5, L133: These comments are unfair and unnecessary. First, the taxonomy we used was the most modern one available and arguably every study (including this one) can be argued to be based on an outdated taxonomy because updates are continuously occurring. Second, the dating errors (which were generally very small; and 25 nodes in bats are also not that many of the total number) were due to a bug in one of the programs, were recognised, and corrected and only the data from the corrected version are now available. There is no need to discredit all the dates in the tree when similar and unknown bugs probably exist in all software used in such analyses, including the ones in this study.

PG 7, L171: I have no idea what "steps of merging, collapsing, and re-scaling" refers to and why this is so bad. Merging and re-scaling would also seem to be part and parcel of the backbone-and-patch method used here, which, albeit in a cruder fashion, was also employed to generate the topology of the mammal supertree. As for the mammal supertree at least, no re-scaling was performed either (also PG 29, L872; the only thing that was fixed was the topology, not the temporal backbone (whatever that is)). The entire topology was dated in one step in one piece.

PG 23, L679: As a personal observation without any real evidence to back it up, my experience has been that dating ancestral nodes based on descendant divergence times (which is what tip-dating essentially does) tends to overinflated the ages of the deepest nodes. In mentioning this to Tanja Stadler, she has also noticed the same phenomenon.

PG 28, L837: Retaining uncertainty has less to do with the backbone-and-patch framework than the Bayesian perspective in which it is performed. Backbone-and-patch (i.e., Mischler's compartmentalization) is arguably also the method that was used to construct the mammal supertree.

PG 28, L853: Not "best considered", they *are* soft polytomies and nothing to the contrary was ever implied.

PG 28, L856: Not supertree methods, but any tress produced in a non-Bayesian framework. Even ML trees mask topological uncertainty. The reason that so few / no polytomies are found there is because it's almost impossible to get equally likely topologies down to the xth decimal place.

PG 28, L869: I would actually argue that deeper dates are more reliable that ones closer to the tips because of the greater confidence in the calibration points and the greater margin of error allowed. It is far easier to confidently assert a fossil as being a mammal than belonging to a given genus, for instance. Genera as such are taxonomically more unstable (cf. Rattus) and, as an unsupported example, a mouse-like fossil will often end up in the genus Mus by default. (Or at least it used to. But how many of these old assertions have been retested?). It's definitely a mammal, but is it Mus or how exactly is it related to Mus? Tricky. Also, Given that mammals are somewhere around 166 million years old, a few million years of uncertainty in the fossil WRT its placement or age won't be a big deal. It definitely can have a major impact when trying to calibrate a node that itself is only a few million years old!

Olaf R.P. Bininda-Emonds

Reviewer #3:

The phylogeny (or rather set of phylogenies) that you have produced here is a big scientific achievement and I’m sure will be a major resource for the community of comparative biologists for years to come. This is a really substantial contribution to the field and I recognize how much effort this represents. Fantastic work! I have one major comment and a couple of very minor comments that I think might improve the paper.

Major comment:

I understand that the main text of the present paper is essentially a reformatted/restructured version of the supplemental material of your previously submitted paper. But I think that broadly speaking, you could do much more to trim the main text more to work better as a paper -- one that can be read and digested by the broad group of people who are likely interested in using this phylogeny. While I appreciate your attention to detail, I think it is worth giving a long hard look at each section and paragraph and deciding whether it is indeed essential for the general reader of this paper or whether it fits better in a supplement. Even as a reviewer (and someone interested in the technical bits of the paper) it was a difficult paper to read as really important methodological points were mixed in with very minute details. Rather than being prescriptive (e.g., keep such and such paragraph in the main text), I am just leaving this as a general impression; please do what you think is best here.

Minor comments:

Perhaps I missed it in the text (and I apologize if I did), but I would ask you to please make available the *dated* set of trees with and without PASTIS-placed species. Some researchers (myself included) who are interested in investigating the diversification dynamics of mammals might prefer to analyze this data without having the distribution of branch lengths assumed by a particular tree model. I know you are providing the ML estimate of the gene-only tree but I was wondering if you were supplying the full dated posterior estimates as well.

First lines of abstract and introduction: I found the focus on speciation/extinction rates a bit misplaced. Honestly proper the largest use case for this set of phylogenies is going to be from comparative biologists who want to use a tree to do PGLS. I think it is worth broadening the opening to discuss some of the important questions in mammalian evolution (including of course diversification rates) that a robust phylogenetic hypothesis can be used to address.

Line 88: I think some credit for the development of this approach to building megaphylogenies should be given to Smith, Beaulieu and Donoghue 2009 BMC https://bmcevolbiol.biomedcentral.com/articles/10.1186/1471-2148-9-37

Line 328: The argument for not using ASTRAL III presented here isn’t very satisfying. I am not suggesting you need to run this program but to say that you didn’t use it because it was released in May 2018 (over a year before you submitted the current version of the paper) seems odd. If it really was a great solution (again, not saying it is) you had more than a year to run the analyses!

Line 604: Minor technical comment (beyond the scope of this paper but perhaps worth mentioning): I think the interpretation of the DR statistic might be improved by formally deriving it in the context of the “identifiable” variables recently described by Louca and Pennell (BioRxiv) https://www.biorxiv.org/content/10.1101/719435v1

Line 652: it is not really clear why using a single tree is likely to lead to overestimation errors (Type 1); what about Type II errors if the single tree is not the same as the posterior mode/median. 

Line 1033: what is the basis for making the recommendation that 100 or 1000 trees will be sufficient to capture uncertainty. I generally think this is a reasonable suggestion but I don’t think you have provided evidence that this is actually a good rule of thumb.

COMMENTS FROM THE ACADEMIC EDITOR:

I think the paper is an improvement and a good fit for the Methods & Resources section.

I agree that the remaining requests are largely textual in nature and should be dealt with through a Minor Revision. I'm not even sure I entirely agree with Rev #3's major comment about the level of detail being problematic given the article type. Sure the paper is dense, but I think people can skim through the sections that are most relevant to them - though maybe his point is more about the level of detail being inconsistent within sections. Irrespective, I do like the rigour and prescription with which steps are presented. 

I did, however, think that Rev #2's major comment needed addressing, especially the point that "ignoring" or "hiding" uncertainty is not inherent to supertrees but rather a reflection of the era in which many of these non-Bayesian methods were developed. This fix should be straightforward. 

The only other point I picked up on, and shared with Rev #3, was the argument for not using ASTRAL III on Line 328 wasn't satisfying (to say the least!).

---

## [Editor Report · Decision Letter 2]

24 Oct 2019

Dear Dr Upham,

On behalf of my colleagues and the Academic Editor, Andrew J Tanentzap, I am pleased to inform you that we will be delighted to publish your Methods and Resources in PLOS Biology. 

Early Version

PRESS 

Kind regards,

Sofia Vickers

Senior Publications Assistant

PLOS Biology

On behalf of, 

Roland Roberts,

Senior Editor

PLOS Biology